# The quality and complexity of pairwise maximum entropy models for large cortical populations

**Valdemar Kargård Olsen**[1], **Jonathan R. Whitlock**[1], **Yasser Roudi**[1,2] *

**1** Kavli Institute for Systems Neuroscience, Faculty of Medicine and Health Sciences, Norwegian University of Science and Technology, Trondheim, Norway, **2** Department of Mathematics, King's College London, London, United Kingdom

* yasser.roudi@ntnu.no

**Data Availability Statement:** The data used in this paper are publicly available and can be downloaded from https://figshare.com/articles/dataset/Rat_3D_Tracking_E-Phys_KISN_2020_Dataset/17903834

## Abstract

We investigate the ability of the pairwise maximum entropy (PME) model to describe the spiking activity of large populations of neurons recorded from the visual, auditory, motor, and somatosensory cortices. To quantify this performance, we use (1) Kullback-Leibler (KL) divergences, (2) the extent to which the pairwise model predicts third-order correlations, and (3) its ability to predict the probability that multiple neurons are simultaneously active. We compare these with the performance of a model with independent neurons and study the relationship between the different performance measures, while varying the population size, mean firing rate of the chosen population, and the bin size used for binarizing the data. We confirm the previously reported excellent performance of the PME model for small population sizes $N < 20$. But we also find that larger mean firing rates and bin sizes generally decreases performance. The performance for larger populations were generally not as good. For large populations, pairwise models may be good in terms of predicting third-order correlations and the probability of multiple neurons being active, but still significantly worse than small populations in terms of their improvement over the independent model in KL-divergence. We show that these results are independent of the cortical area and of whether approximate methods or Boltzmann learning are used for inferring the pairwise couplings. We compared the scaling of the inferred couplings with $N$ and find it to be well explained by the Sherrington-Kirkpatrick (SK) model, whose strong coupling regime shows a complex phase with many metastable states. We find that, up to the maximum population size studied here, the fitted PME model remains outside its complex phase. However, the standard deviation of the couplings compared to their mean increases, and the model gets closer to the boundary of the complex phase as the population size grows.

## Author summary

With recent major advances in recording technology, much of computational neuroscience has effectively turned into describing patterns in large amounts of data as succinctly as possible. One way to do this is to construct simple parametric models of the probability

All code written in support of this publication is publicly available at https://osf.io/dajc6/.

**Funding:** Research Council of Norway Centre of Neural Computation, grant number 223262; Research Council of Norway NORBRAIN, grant number 295721; The Kavli Foundation. The funders had no role in study design, data collection and analysis, decision to publish, or preparation of the manuscript.

**Competing interests:** The authors have declared that no competing interests exist.

distribution over patterns of neuronal activity, such as the pairwise maximum entropy model. Intuitively, the pairwise model makes the distribution over all patterns as flat or uniform as possible, while keeping all firing rates and pairwise correlations the same as in the data. This model has been shown to effectively capture the observed distribution of activity patterns well for small populations ($\sim 10$), but it has not been systematically studied for large populations. Here, we study the performance of the pairwise model using a Neuropixel dataset recorded from the visual, auditory, somatosensory, and motor cortices of freely moving rats exposed to different stimuli. Consistent with previous findings, we find good performance for small populations, before it falls sharply as the population size increases ($\gtrsim 25$). However, we also find that this decrease in performance reveals interesting differences between the correlation structure of the data recorded under different sensory conditions.

## Introduction

Neuronal activity in the brain is correlated and stochastic. These correlations can be at the level of spikes or firing rates, and can vary in response to behavior, stimulus, or brain state [1]. But what is a good probabilistic model of the correlated activity of neurons? This has been an important long-standing question, as answering it allows us to describe neural activity in simpler terms [2], and it is crucial to understanding the computations that populations of neurons perform [3–5].

Generally speaking, the problem of which probabilistic model should be used to describe an event involves a tension between (a) the simplicity of the model, often reflected in its number of parameters, and (b) the model's precision in describing the probabilistic events. To these, one may also add two practical conditions: (c) easy inference of the parameters and (d) the ease with which the model can be sampled from.

Binned with time bins of size $\delta t$, the spiking activity of a neuron $i$ in the time bin $t$, can be described by a binary (spin) variable $s_i(t)$, taking the value 1 if the neuron fires and $-1$ otherwise [6–10]. The problem of describing probabilistic correlated neural activity then involves writing down $p_t(\mathbf{s})$ where $\mathbf{s}(t) = (s_1(t), s_2(t), \cdots, s_N(t))$. Ignoring temporal dynamics by assuming stationarity leads to the even simpler problem of writing down $p(\mathbf{s})$. If the neurons were independent, this would be a simple model to fit to the data. However, when there are correlations, even this distribution requires estimation of the probability of $2^N$ states. Therefore, we reduce the dimensionality of this problem even further, while still not ignoring all correlations, by using the maximum entropy principle [11]. Limiting oneself to the means and equal-time pairwise correlations, one then obtains the so-called pairwise maximum entropy (PME) model/distribution. With reference to the four conditions above, the model is good in terms of having a small number of parameters ($N(N + 1)/2$ compared to $2^N$; condition a). Although inferring these parameters exactly requires time-consuming Boltzmann learning, efficient and fast methods for inference exist [9, 12–19] (condition c), thanks largely to the mathematical similarity between a fitted PME model and the Sherrington-Kirkpatrick (SK) model [20–22].

In terms of performance (condition b), the results are, however, inconclusive: different studies have used different measures to evaluate the performance of the model and on data from different areas of the brain and with different population sizes. An important measure of performance is the Kullback-Leibler (KL) divergence between the data distribution and the pairwise model, compared to the KL-divergence between the data distribution and an independent model [6, 7, 10]. According to this measure, the PME model is found to be almost

perfect for small populations ($N \sim 10$) of retinal neurons [6, 7, 23]. Similarly promising results have been found in the cortex [24–27], but still for small populations. Studies on larger populations used other performance measures or modified versions of the model [8, 23, 28–31]. Among these, Tkacik et. al. [30] added the probability of $m$ neurons being simultaneously active as an additional constraint to the model for $N \sim 100$ retinal neurons. Shimazaki et. al. [28] showed that a simple way to include higher-order correlations is to add the probability that all neurons are silent as an additional constraint, and observed good performance for CA3 activity recorded using calcium imaging. For a sufficiently homogeneous population of size $N$ and average firing rate $\bar{v}$ binarized at bin size $\delta t$, Roudi et. al. [10] used a perturbative expansion in $N\bar{v}\delta t$ to derive an analytic expression for the performance of the PME model. They showed, and it was also confirmed using simulated data [9, 14], that when $N\bar{v}\delta t << 1$, the pairwise model is always much better than the independent model in terms of KL-divergence. This leads to excellent performance regardless of whether the true distribution of the data is pairwise or not. In addition, this performance should decay linearly with $N\bar{v}\delta t$ as long as $N\bar{v}\delta t << 1$. However, the performance in this perturbative regime is not predictive of the large $N$ behavior, which is our primary interest.

Much less attention has been paid to the ease with which the inferred model can be sampled from (condition d above). Probabilistic models, such as the SK model, can be in a complex/ hard phase that makes sampling from them difficult [32]. This may happen when the number of metastable states, where sampling algorithms can get stuck in, grows (often exponentially) for large $N$ [22, 33, 34]. A simple way to measure this complexity is to consider the ratio of the standard deviation of the couplings to their mean. This has only been studied in the context of PME models and neural populations in the case of simulated data [9], where extrapolation to realistic population sizes is not meaningful.

In this paper, we report a comprehensive study of the performance of the PME model for large cortical populations. We analyzed spike trains of up to 100 simultaneously recorded neurons from the visual, auditory, somatosensory, and motor cortices of rats that freely forage for food in an open arena. We evaluated the performance of the pairwise model based on its KL-divergence with data and compared it to the independent model. We also measured the performance of the PME model in predicting third-order correlations in the data and the distribution of $m$ simultaneously active neurons. We studied the relationship between these measures and also the relationship between model quality and $N$, $\bar{v}$, and $\delta t$, in addition to $N\bar{v}\delta t$. Finally, we evaluated the complexity of the PME model for large populations by studying the scaling of inferred couplings with $N$ and the analogy with the SK model.

We find that the PME model is an excellent model of these cortical populations in terms of KL-divergences only for small $N$. Small $\delta t$ and/or small $\bar{v}$ also increase performance, but reducing them cannot make the performance for large populations as good as that for small populations. Performance rapidly declines as these quantities increase, and this happens regardless of whether the analysis is restricted to data from a single brain region or experimental condition, or whether the data is pooled between regions and experimental conditions. Differences in performance between different populations and different experimental conditions can be used to compare the role of correlations in shaping neural activity [35]. Exemplifying this, we divided the neural data recorded from the visual cortex into lights-on and lights-off periods. We find a better performance in the lights-on condition than in the lights-off condition. This was primarily because, in the lights-off condition, the independent model was a better model for the neural data. This phenomenon appears to be specific to the visual cortex, as such differences were absent in auditory cortex between sound-on and sound-off conditions. A second potential use case of the PME model that we demonstrate is that they can be used to assign informative and stable effective connectivity between neurons [9]. Finally, we show that

although the PME model's ability to predict third-order correlations and the probability of simultaneous spikes also decreases with $N$, it does well in predicting them in a range of $N$ exceeding the range of good performance according to the KL-divergence.

Regarding the scaling of the couplings, we find that the inferred couplings and the mean-field theory of the SK model exhibit a similar dependence on $N$. The standard deviation of the couplings is substantially greater than the mean, and the ratio of the standard deviation to the mean increases with $N$, reaching a value $\sim 10$ for $N = 490$. However, we find that the model still remains in its normal phase, but also that this phase approaches its instability as $N$ grows. This is suggestive of a large number of metastable states, making the PME model more and more difficult to properly sample from as $N$ grows.

## Materials and methods

### The dataset

We use a publicly available Neuropixel data set recorded from the visual, auditory, somatosensory, and motor cortices of freely moving rats [36, 37]. Each $\sim 20$ minute session consisted of the rat foraging in an octagonal ($2 \times 2 \times 0.8$ m) arena in dim light, in darkness, with a small weight attached to the implant, or with random interval white noise playing. Here, we mainly consider the neurons shared across six such sessions (2 light, 2 dark, 1 weight, 1 noise) recorded from the same probe in the same animal on the same day. This results in approximately 2 hours of data from $N = 495$ neurons, 130 of which are from auditory cortices, and the remaining 365 neurons from visual cortices. These sessions were concatenated and binned at $\delta t$ seconds. When not stated otherwise, a bin size of $\delta t = 0.02$ seconds is used, giving $\sim 450,000$ samples.

The performance of the pairwise model in the visual, auditory, somatosensory, and motor cortices will also be compared. For this, we used data from four sessions (1 light, 1 dark, 1 weight, 1 noise) recorded from the same probe in the same animal on the same day. The neurons shared across these four sessions, recorded for approximately 1 hour and 20 minutes, were concatenated and binned as above. This resulted in $N = 539$ neurons from visual cortices, $N = 376$ from auditory cortices, $N = 287$ from somatosensory cortices, and $N = 1115$ from motor cortex. Although data from the four different experimental conditions have been analyzed mainly together, we also investigated the effect of room lighting on visual cortex neurons and the effect of white noise on auditory cortex neurons. This allows us to assess whether changing sensory input impacts correlations among active neurons for each modality.

### The pairwise maximum entropy model

We consider data from $N$ neurons and binned the spikes in time bins of size $\delta t$. The state of neuron $i$ in time bin $t$ is then represented by a binary spin variable $s_i(t)$ which is equal to $+1$ if neuron $i$ spikes at least once in that time bin, and $-1$ otherwise. The data are thus represented by a binary vector of length $N$, $\mathbf{s}(t) = [s_1(t), s_2(t), \ldots, s_N(t)]$, for each time bin $t = 1 \cdots T$. We define the means and correlations we want the model to conserve as

$$\langle s_i \rangle \equiv \frac{1}{T} \sum_{t=1}^{T} s_i(t), \quad \langle s_i s_j \rangle \equiv \frac{1}{T} \sum_{t=1}^{T} s_i(t) s_j(t). \tag{1a, b}$$

The pairwise model is then given by

$$p_{\text{pair}}(\mathbf{s}) \equiv \frac{1}{Z_{\text{pair}}} \exp\left( \sum_i h_i s_i + \sum_{i<j} J_{ij} s_i s_j \right), \tag{2}$$

and its parameters $h_i$ and $J_{ij}$ are chosen such that $\langle s_i \rangle_{\text{pair}} = \langle s_i \rangle$ and $\langle s_i s_j \rangle_{\text{pair}} = \langle s_i s_j \rangle$, where $\langle \cdots \rangle_{\text{pair}}$ represents averages with respect to the distribution in Eq (2). Note that the couplings are symmetric ($J_{ij} = J_{ji}$) and that self-connections are omitted ($J_{ii} = 0$), resulting in a total of $N(N+1)/2$ parameters. The biases $h_i$ and couplings $J_{ij}$ can be found using Boltzmann learning [38], or approximate methods such as pseudo-likelihood maximization [15, 19, 39, 40].

The pseudo-likelihood approximation [39] decomposes the problem of finding biases and couplings into $N$ independent sub-problems considering the conditional distribution of each neuron $s_i$ given the set of all other neurons, $s_{/i}$:

$$p(s_i | s_{/i}) = \frac{\exp\left(s_i\left[h_i + \sum_{j \neq i} J_{ij} s_j\right]\right)}{2 \cosh\left(h_i + \sum_{j \neq i} J_{ij} s_j\right)} = \frac{1}{1 + \exp\left(-2 s_i\left[h_i + \sum_{j \neq i} J_{ij} s_i\right]\right)}. \tag{3}$$

The sum of these conditional distributions replace the likelihood function and is maximised over $h_i$ and $J_{ij}$. Equivalently, the conditional distributions define $N$ independent logistic regression problems, each resulting in an $h_i$ (the zeroth coefficient) and a row $i$ in the coupling matrix $J$. Because this results in an asymmetric coupling matrix $J$, we follow the suggestion of [19] and use the average $\frac{1}{2}(J_{ij} + J_{ji})$ as our couplings. This approximation has been shown to estimate the parameters obtained from Boltzmann learning well [15, 19, 40, 41; S6 Fig]. Unless otherwise stated, pseudo-likelihood has been used to approximate $h$ and $J$.

## Assessing performance: KL-divergence and entropy differences

To assess the quality of the pairwise model, we used several measures inspired by and used in previous studies. Consider the KL-divergence between the pairwise and the true distribution:

$$d_{\text{pair}} \equiv D_{\text{KL}}(p_{\text{true}} \| p_{\text{pair}}) = \sum_{\mathbf{s}} p_{\text{true}}(\mathbf{s}) \ln \frac{p_{\text{true}}(\mathbf{s})}{p_{\text{pair}}(\mathbf{s})} = S_{\text{pair}} - S_{\text{true}} \tag{4}$$

where $S_{\text{true}} = -\sum_{\mathbf{s}} p_{\text{true}}(\mathbf{s}) \log p_{\text{true}}(\mathbf{s})$ is the entropy of the true underlying distribution of the data $S_{\text{pair}} = -\sum_{\mathbf{s}} p_{\text{pair}}(\mathbf{s}) \log p_{\text{pair}}(\mathbf{s})$. The last equality follows from the fact that the cross-entropy term in $d_{\text{pair}}$ is equal to the entropy of the pairwise model fitted to the data:

$$\begin{aligned}
S_{\text{pair}} &= \sum_i \langle s_i \rangle_{\text{pair}} \sum_{i<j} \langle s_i s_j \rangle_{\text{pair}} + \log Z_{\text{pair}} \\
&= \sum_i \sum_{\mathbf{s}} p_{\text{true}}(\mathbf{s}) s_i + \sum_{i<j} p_{\text{true}}(\mathbf{s}) s_i s_j + \log Z_{\text{pair}} \\
&= -\sum_{\mathbf{s}} p_{\text{true}}(\mathbf{s}) \log p_{\text{pair}}(\mathbf{s}),
\end{aligned} \tag{5}$$

where we have used the fact that for the pairwise fit $\sum_s p_{\text{true}}(\mathbf{s}) s_i s_j = \sum_s p_{\text{pair}}(\mathbf{s}) s_i s_j$ and $\sum_{\mathbf{s}} p_{\text{true}}(\mathbf{s}) s_i = \sum_s p_{\text{pair}}(\mathbf{s}) s_i$ are satisfied.

Our first measure is based on the KL-divergence $d_{\text{pair}}$, or equivalently as shown above, the entropy difference. The degree to which pairwise correlations explain the data can be evaluated by comparing $d_{\text{pair}}$ with an independent maximum entropy model. In this case, for any given bin, the probability that a neuron $i$ spikes according to the model is independent of other neurons, while matching $\langle s_i \rangle$. As such, the joint probability of observing $\mathbf{s}$ is

$$p_{\text{ind}}(\mathbf{s}) \equiv \prod_i p_i(s_i) = \prod_i \frac{\exp[h_i^{\text{ind}} s_i]}{2 \log \cosh[h_i^{\text{ind}}]} = \frac{1}{Z_{\text{ind}}} \exp\left(\sum_i h_i^{\text{ind}} s_i\right), \tag{6}$$

where $h_i^{\text{ind}} = \tanh^{-1} \langle s_i \rangle_{\text{true}}$. This comparison quantifies how much of the correlation structure

in the data is accounted for by pairwise correlations. In fact, following previous work [6, 7, 9, 10], we can define a goodness-of-fit measure as

$$G \equiv 1 - \frac{d_{\text{pair}}}{d_{\text{ind}}} = \frac{S_{\text{ind}} - S_{\text{pair}}}{S_{\text{ind}} - S_{\text{true}}} \qquad (7)$$

If $G = 1$, then the pairwise model and the true distribution are identical. On the other hand, if $G \sim 0$ the pairwise model is as good as the independent model. Consequently, one is not gaining much by including pairwise correlations.

Estimating $G$ for large populations is difficult as it requires estimating the entropy of the data and the partition function $Z_{\text{pair}}$ of the pairwise model. When the number of neurons is small ($N \lesssim 20$), the partition function can be computed exactly by summing over all states. For larger population sizes, one needs to appeal to approximations. Therefore, in the Results section, we analyze the cases of $N \leq 20$ and $N \geq 20$ separately. For the latter case, we describe an approximation that we demonstrate to estimate $Z_{\text{pair}}$ and $G$ very well. In what follows, we use $\hat{Z}_{\text{pair}}$ and $\hat{G}$ when referring to these approximations.

## Assessing performance: Predicting third order correlations and the distribution of $m$ active neurons

In addition to the measures based on KL-divergences mentioned in the previous subsection, the performance of the pairwise model could also be evaluated by comparing the (connected) third-order correlations in the data with those predicted from the inferred pairwise model [30, 31, 42, 43]. Third-order correlations $C_{ijk}$ are defined as

$$C_{ijk} \equiv \langle s_i s_j s_k \rangle. \qquad (8)$$

While some studies use this directly [42, 43], others [30] consider connected third-order correlations:

$$\widetilde{C}_{ijk} \equiv \langle (s_i - \langle s_i \rangle)(s_j - \langle s_j \rangle)(s_k - \langle s_k \rangle) \rangle. \qquad (9)$$

Like for the KL-divergences, we can define measures of goodness $G_C$ and $G_{\widetilde{C}}$ based on the third-order and connected third-order correlations, by comparing the pairwise and independent model as follows:

$$G_C \equiv 1 - \sqrt{\frac{\sum_{i<j<k}(C_{ijk}^{\text{data}} - C_{ijk}^{\text{pair}})^2}{\sum_{i<j<k}(C_{ijk}^{\text{data}} - C_{ijk}^{\text{ind}})^2}}, \quad G_{\widetilde{C}} \equiv 1 - \sqrt{\frac{\sum_{i<j<k}(\widetilde{C}_{ijk}^{\text{data}} - \widetilde{C}_{ijk}^{\text{pair}})^2}{\sum_{i<j<k}(\widetilde{C}_{ijk}^{\text{data}} - \widetilde{C}_{ijk}^{\text{ind}})^2}} \qquad (10a, b)$$

Here, $C_{ijk}^{\text{data}}$ and $\widetilde{C}_{ijk}^{\text{data}}$ are calculated from the data, $C_{ijk}^{\text{pair}}$ and $\widetilde{C}_{ijk}^{\text{pair}}$ are calculated from samples of the inferred pairwise model, and $C_{ijk}^{\text{ind}}$ and $\widetilde{C}_{ijk}^{\text{ind}}$ are calculated from samples of the inferred independent model.

Finally, in addition to KL-divergences and third-order correlations, the performance of the pairwise model has also been evaluated by how well it predicts $H_m$, the probability that $m$ neurons are simultaneously active in a time bin [23, 29–31, 42–44]. Like for the third-order correlations, we define a measure that compares the predictions made by the pairwise and independent model:

$$G_H \equiv 1 - \sqrt{\frac{\sum_m(H_m^{\text{data}} - H_m^{\text{pair}})^2}{\sum_m(H_m^{\text{data}} - H_m^{\text{ind}})^2}}. \qquad (11)$$

Here, $H_m^{\text{data}}$ is calculated from the data, $H_m^{\text{pair}}$ is calculated from samples of the inferred pairwise model, and $H_m^{\text{ind}}$ is calculated from samples of the inferred independent model.

## Results

In what follows, we study the performance of the pairwise model according to the measures described in the Methods section. When not stated otherwise, we pick neurons from our longest dataset consisting of 130 neurons from the auditory cortices and 365 neurons from visual cortices, recorded for six sessions.

Previous work [9, 10] has shown that for a population of size $N$, consisting of neurons with a firing rate of $v_i$, and binned with a time bin of $\delta t$, the quantity $G$ in Eq (7) can be written as

$$G \approx 1 - \frac{g_{\text{pair}}}{g_{\text{ind}}} N \bar{v} \delta t + \mathcal{O}(N \bar{v} \delta t)^2 \qquad (12)$$

where $\bar{v} = \frac{1}{N} \sum_i v_i$ and where $g_{\text{pair}}$ and $g_{\text{ind}}$ do not depend on $\delta t$ or $N$ [10]. This can be intuitively understood by noting that the main contribution to $d_{\text{pair}}$ comes from third-order correlations of which there are $\mathcal{O}(N^3)$ and which, for small $\delta t$ scale as $(\bar{v} \delta t)^3$ so $d_{\text{pair}} \sim (N \bar{v} \delta t)^3$. On the other hand, the main contributions to $d_{\text{ind}}$ come from pairwise correlations, of which there are $\mathcal{O}(N^2)$ and their size scale as $(v \delta t)^2$, which yields $d_{\text{ind}} \sim (N \bar{v} \delta t)^2$ and the leading order in Eq (12).

Eq 12 implies that for $N \bar{v} \delta t \ll 1$, $G \approx 1$ and the pairwise model is guaranteed to work well. The analysis, however, does not predict what happens if $N \bar{v} \delta t$ is not small. Furthermore, in practice, different populations with the same size and average firing rate may have different values of $g_{\text{pair}}$ and $g_{\text{ind}}$, depending on the degree of heterogeneity of neural spiking in the population. In what follows, then, we evaluate the performance of the pairwise model as a function of $N \bar{v} \delta t$, for both small values ($< 1$) and larger values. This gives us a natural way to compare the effect of not only varying the population size (as was done on simulated data in [9]), but also of using different bin sizes and selecting populations of the same size but different average firing rates. We find that $N$ affects $G$ more than $\bar{v}$ and $\delta t$, therefore we show our main results as a function of $N$ in addition to $N \bar{v} \delta t$.

### $N \leq 20$ and exact $Z_{\text{pair}}$

To calculate the performance $G$ we first need to know the true distribution $p_{\text{true}}$ that would emerge if we had infinite data. Of course, infinite data is unavailable, so $p_{\text{true}}$ is often taken to be the frequency of each activity pattern $\mathbf{s}$ in the data, denoted by $p_{\text{data}}$. In S2 Appendix we show that the finite sampling bias resulting from this assumption does not substantially affect our results. Second, we need to know the probability of all sampled states $\hat{\mathbf{s}}$ according to the pairwise model, which requires the partition function $Z_{\text{pair}}$. For small populations, one can calculate $Z_{\text{pair}}$ exactly, which we do here for up to $N = 20$. While we can only calculate $Z_{\text{pair}}$ exactly for a small $N$, we can do it for arbitrary bin sizes $\delta t$ and mean firing rates $\bar{v}$.

Fig 1A and 1B shows the $N$ dependence of entropies and KL-divergences for populations of up to 20 neurons. These plots show a trend similar to those in [9] for simulated data from a balanced excitatory-inhibitory network: the data entropy and those of the independent and pairwise models are very close to each other (Fig 1A) and their differences, as reflected in the KL-divergences, are very small. However, these differences increase with $N$, with $d_{\text{ind}}$ increasing faster than $d_{\text{pair}}$. Fig 1B shows how $G$ changes with $N$: as predicted by the perturbative expansion [10], for small $N$, there is a linear decay, followed by a further drop in $G$. Compared to the results of the simulated data in [45], $G$ drops faster in this dataset, and the linear (perturbative) regime seems to be smaller. This can be explained by the larger bin size used in this

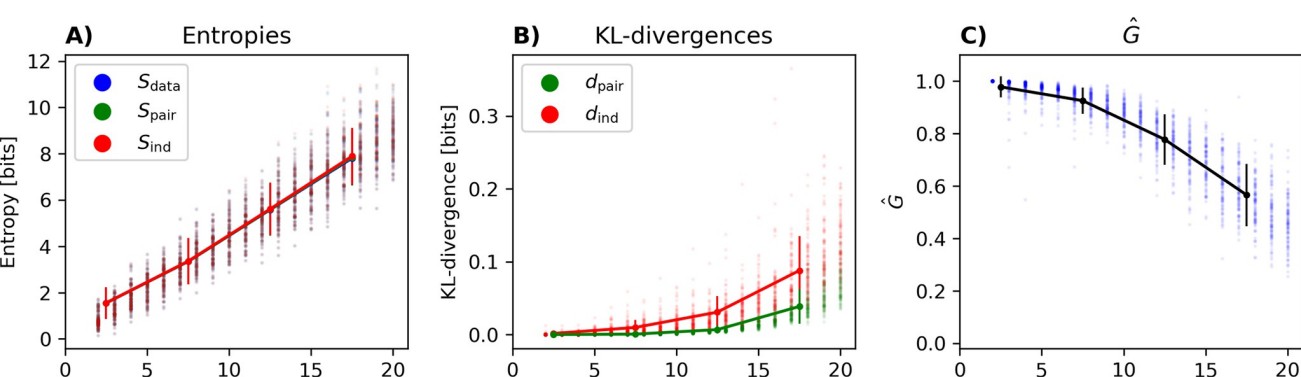

**Fig 1. Model properties versus *N* for small *N*.** (A) The entropy of the data $S_{\text{data}}$, independent model $S_{\text{ind}}$, and the PME model $S_{\text{ind}}$, (B) the KL-divergences $d_{\text{ind}}$ and $d_{\text{pair}}$, and (C) *G*, all plotted versus *N* for populations of up to 20 neurons. Each dot represents one population. For each *N*, we have run the analysis for 100 randomly selected populations. The solid lines are connect points found by connected averages of the quantities over populations in ranges $N = 2 - 5, 5 - 10$ etc. and error bars are the standard deviations.

paper (thus larger $N\bar{v}\delta t$), as well as the simulated data in [45] being from a balanced network with, most likely, weaker correlations between neurons than here.

We also studied in more detail how the entropies, the KL-divergences, and the performance measure *G* vary with $N\bar{v}\delta t$ as we change *N*, $\bar{v}$ and $\delta t$ separately. To evaluate the effect of population size, we fixed $\delta t = 0.02$ seconds and selected 100 populations of $N = 2, 3, \ldots, 20$ neurons randomly from the 495 neurons in our dataset. To evaluate the effect of bin size, we randomly choose 5000 populations of size $N = 20$ and picked a bin size $\delta t$ uniformly between 0.005 and 0.2 seconds. Evaluating the effect of $\bar{v}$ is less straightforward as it requires us to select non-random populations out of the 495 neurons that cover a wide range of $\bar{v}$ and thus $N\bar{v}\delta t$. Therefore, we choose each population by picking a neuron *i* with a probability proportional to its firing rate and then choosing the remaining neurons *j* with a firing rate similar to the first one. That is, the remaining neurons *j* were picked (without replacement) with probability $p(j) = \frac{1}{|v_i - v_j|^3} / \sum_k^N \frac{1}{|v_i - v_k|^3}$, where the exponent controls the spread of the firing rates within each population. This results in sub-populations that spread out nicely along the range of possible mean firing rates. Of course, by construction, these populations are not representative samples of neurons in the dataset, but they allow us to look at the performance measure for real populations with different mean firing rates.

In Fig 2A–2C, we first show the entropies of the data, pairwise model and independent model versus $N\bar{v}\delta t$, changing *N*, $\bar{v}$, and $\delta t$ as described above. Note that Fig 2A, 2D and 2G is the same as Fig 1 except that the points are not grouped by *N*. The results for the KL-divergences can be seen in Fig 2D–2F, where one observes that the distance between the independent model and the data $d_{\text{ind}}$ increases more rapidly with $N\bar{v}\delta t$ than the distance between the pairwise model and the data $d_{\text{pair}}$. In both Fig 2B and 2E, we observe a branching of the entropy and KL-divergences as $\bar{v}$ increases. We shall get back to the origin of this in more detail below.

Fig 2G–2I show the performance of the pairwise model as quantified by *G* versus $N\bar{v}\delta t$. We see in Fig 2G that *G* decays rapidly for large populations, not only when plotted versus *N* itself as in Fig 1C, but also when considered as a function of $N\bar{v}\delta t$. The initial decline shown here in *G* with $N\bar{v}\delta t$ is consistent with the prediction of the perturbative expansion in $N\bar{v}\delta t \ll 1$ [10],

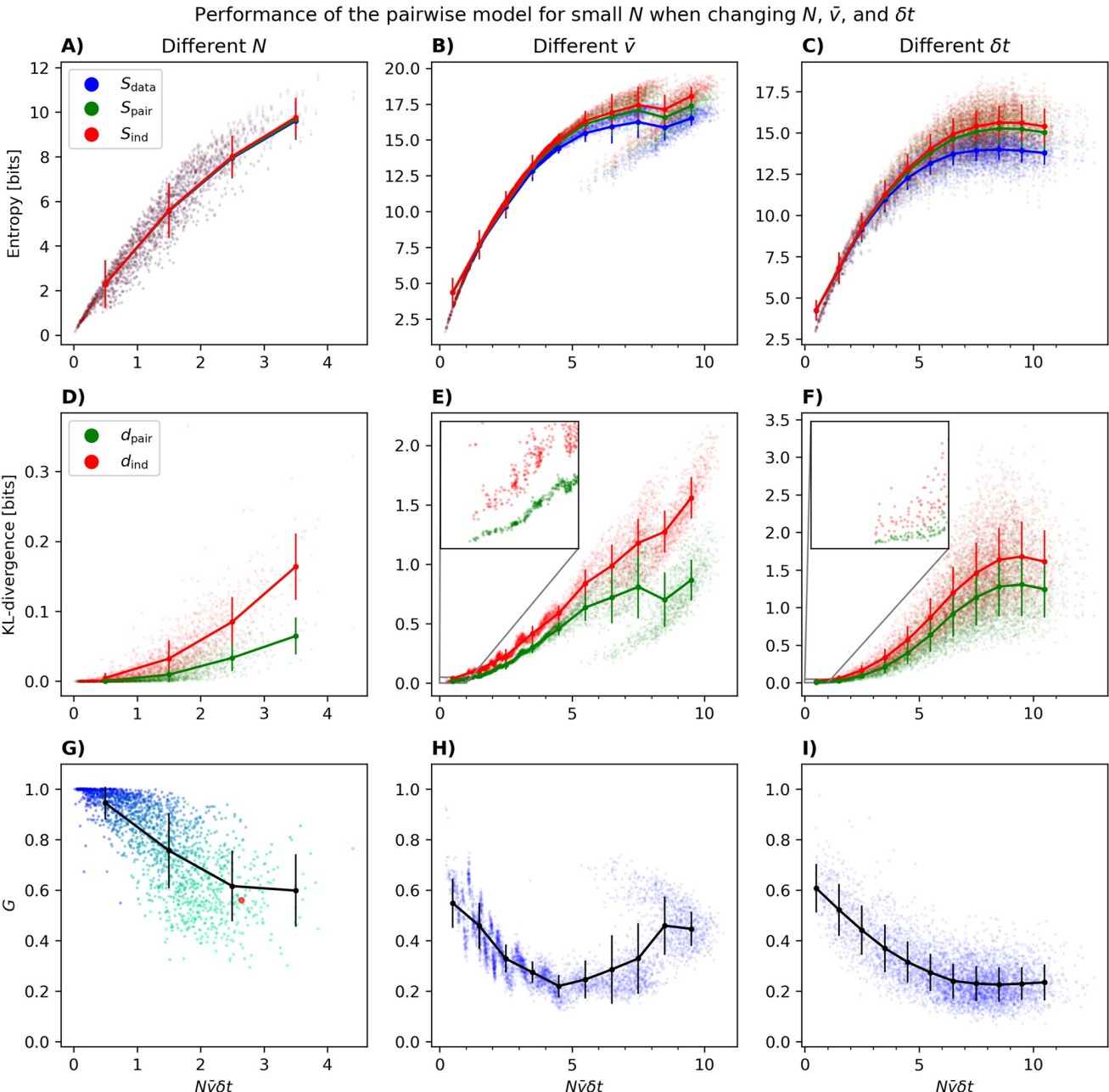

**Fig 2. Model properties versus the perturbation parameter $N\bar{v}\delta t$ for small $N$.** (A—C) The entropies of the data and the models versus $N\bar{v}\delta t$ when this parameter is changed by changing $N$, $\bar{v}$ and $\delta t$ respectively. (D—F) same as (A—C) but for the KL-divergences $d_{ind}$ and $d_{pair}$. The insets show the zoomed versions for $N\bar{v}\delta t = 0 - 1$ and $d_{ind}/d_{pair} = 0 - 0.05$. (G—I) same as above but for $G$ versus $N\bar{v}\delta t$. When changing N (first column), 100 populations of size $N$ were chosen randomly, for each $N$ between 2 and 20. When changing $\bar{v}$ (second column), 5000 populations of size $N = 20$ were chosen so that neurons have similar firing rates (see text). In panel G the color gradient represents the number of neurons in the population (blue [few] → turquoise [many]). When changing $\delta t$ (third column), 5000 populations consisting of $N = 20$ neurons were chosen and binned with a binsize chosen uniformly between 0.005 and 0.2 seconds. Panels A, D, and G contain the same data as in Fig 1 except that in the latter, populations of the same size are lumped together.

but our results here show that the decay continues well outside the perturbative regime. Fig 2H, shows that for $N = 20$ and $\delta t = 0.02$ seconds, $G$ is well below 1 even for the smallest $\bar{v}$. We also observe an initial drop until we see an increase again. The root of this increase is the same as the branching of the entropy and KL-divergences for larger $\bar{v}$ in Fig 2B and 2E.

To better understand the nature of this increase, and given that we had to select non-random populations to obtain a wide range of $\bar{v}\delta t$, we tested two hypotheses: (1) that the large firing rate populations have a different fraction of neurons from auditory versus visual cortex, and (2) that the mean firing rates vary more form one neuron to another, i.e. population heterogeneity. Fig 3 shows that the main effect comes from the change in the standard deviation of the firing rates of the populations. Focusing in particular on the region of $N\bar{v}\delta t = 6 - 8$, it is clear that with the same average firing rate, populations with greater variability in firing of their neurons have larger $G$.

Unlike Fig 2G where small $N\bar{v}\delta t$ is achieved by having a small $N$, in Fig 2H and 2I, $G$ does not become close to 1 for small $N\bar{v}\delta t$. One can understand this result by noting that when $\bar{v}$ or $\delta t$ are small, as is the case for the extreme left parts of Fig 2H and 2I, there is a small chance of seeing any time bin with more than one neuron spiking, making neurons effectively independent, although the pairwise model is still better (note the different ranges of the y-axes in Fig 2G–2I). In other words, in the case of $N = 20$ but small $\delta t$ or small $\bar{v}$, the performance of the pairwise model is not good because the independent model is already a pretty accurate description of the data; this is not true when $N\bar{v}\delta t$ is small for small $N$. Seen differently, for a fixed population size, $G$ can become substantially smaller than 1 in two different ways. First, for a population size $N = 20$, the rightmost points in Fig 2G have similar values of $d_{\mathrm{pair}}$ and $d_{\mathrm{ind}}$ ($\sim 0.05$ and $\sim 0.15$) leading to $G \sim 0.6$. Second, for the same population size, but with smaller $\bar{v}$ and smaller $\delta t$ (leftmost points in Fig 2H and 2I), both $d_{\mathrm{ind}}$ and $d_{\mathrm{pair}}$ are substantially smaller (see insets in Fig 2E and 2F) but again leading to $G \sim 0.6$. In other words, $G$ can be substantially different from 1 because the PME and independent models are equally bad (right part of Fig 2A) or because they are equally good (left part of Fig 2H and 2I). This just reflects the fact that $G$ is the ratio between $d_{\mathrm{ind}}$ and $d_{\mathrm{pair}}$ and that for the same value of $G$ far from 1, both $d_{\mathrm{ind}}$ and $d_{\mathrm{pair}}$ can be relatively large ("equally bad") or relatively small ("equally good"). These two cases, of course, differ in the degree to which adding higher-order interactions improves the model. We demonstrate this by fitting a maximum entropy model with third-order interactions $K_{ijr}$ (in addition to $J_{ij}$ and $h_i$) to a representative population with $N = 15$ neurons. This was done using Boltzmann learning (learning rate of $\eta = 0.01$ and 100000

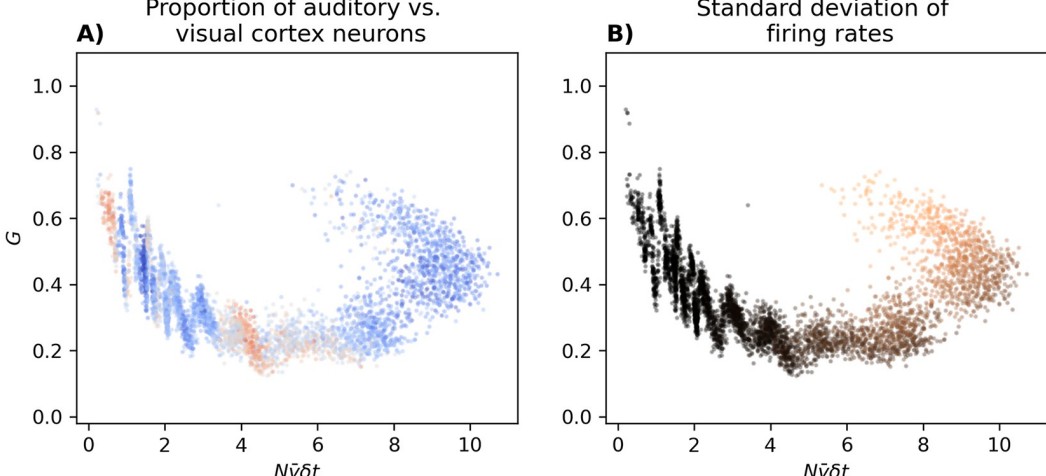

**Fig 3. Effect of proportion of neurons from visual cortex versus auditory cortex in the populations (A; blue [visual] → red [auditory]), and the standard deviation of the firing rates of neurons in the populations (B; black [small] → red [large]) on the dependence of $G$ on $\bar{v}$.** The points are the same as those in Fig 2H, but color coded differently.

iterations). Binned at 0.02 seconds, we have $G = 0.56$ for this population (red dot in Fig 2G). As expected, there is a slight improvement when the data from this population is binned at 0.01 seconds, leading to $G = 0.61$ and a slight decay when binned at 0.08, giving $G = 0.50$. These are both far from 1, but for the different reasons mentioned above: at 0.08, $d_{\text{ind}} = 0.3082$ and $d_{\text{pair}} = 0.1551$ is large compared to $d_{\text{ind}} = 0.0175$ and $d_{\text{pair}} = 0.0068$ for 0.01 seconds. In other words, when binning at 0.01 the independent model is already an excellent model, and adding pairwise interactions offer little improvement. Similarly, adding third-order interactions in the case of 0.01 seconds adds little to the model quality ($d_{\text{triplet}} = 0.0063$). However, at 0.08 seconds, adding third order correlations lead to $d_{\text{triplet}} = 0.1165$, an improvement of 25% over $d_{\text{pair}}$. A similar behaviour was observed in other selected populations.

The results of Fig 2G are also reproduced for $\delta t = 0.01$ seconds in S1 Fig, where we show figures similar to Fig 2H and 2I for $N = 10$. This confirms that the general trends do not depend on these choices. Taken together, the results of this section show that it is for small populations and small values of $N\bar{v}\delta t$, and only in this regime, that $G$ is close to 1 and the pairwise model shows an excellent performance. $G$ generally decrease as $N\bar{v}\delta t$ becomes larger, although for $N = 20$ in Fig 2H there is an eventual increase associated with the larger variability in the firing rates.

## $N \geq 20$ and approximating $Z_{\text{pair}}$

Although the excellent performance of the pairwise model as indicated by $G$ close to unity is only observable for small population sizes and small $N\bar{v}\delta t$, it is still important to note that, for example in Fig 2G, the smallest value of $G$ is $\sim 0.6$, meaning that the pairwise model still offers a substantial improvement over the independent model for such population sizes. Therefore, we sought to quantify this improvement in cortical areas for larger populations.

As above, letting $p_{\text{true}}$ be $p_{\text{data}}$ allows us to estimate the entropy of the data $S_{\text{data}}$; see S2 Appendix. For computing $S_{\text{pair}}$, the first two sums in Eq (5) can be easily performed as they only depend on the means and correlations of the data (second line Eq (5)). However, the term $\log Z_{\text{pair}}$ becomes intractable for large $N$, as the summation over all states in $Z_{\text{pair}}$ cannot be performed. Therefore, we must turn to approximations.

To estimate $Z_{\text{pair}}$, our starting point is that $Z_{\text{pair}} = \exp[-E(\mathbf{s})]/p_{\text{pair}}(\mathbf{s})$ for any given state $\mathbf{s}$, where $E(\mathbf{s}) = -\sum_i h_i s_i - \sum_{i<j} J_{ij} s_i s_j$. Consequently, we have

$$Z_{\text{pair}} = \arg\min_{\alpha} \sum_{\mathbf{s}} \left[ p_{\text{pair}}(\mathbf{s}) - \frac{\exp[-E(\mathbf{s})]}{\alpha} \right]^2. \tag{13}$$

The idea is that one can use $p_{\text{data}}$ instead of $p_{\text{pair}}$ in the above expression and perform the summation only over the set of states that were observed at least once $\hat{\mathbf{s}} \in \mathcal{O}$. Using this approximation, and setting the derivative of Eq (13) equal to 0, yields the following expression as an estimation of the partition function:

$$\hat{Z} \equiv \frac{\sum_{\hat{\mathbf{s}} \in \mathcal{O}} \exp\left[2\sum_i h_i s_i + 2\sum_{i<j} J_{ij} s_i s_j\right]}{\sum_{\hat{\mathbf{s}} \in \mathcal{O}} p_{\text{data}}(\hat{\mathbf{s}}) \exp\left[\sum_i h_i s_i + \sum_{i<j} J_{ij} s_i s_j\right]}. \tag{14}$$

This way of estimating $Z$ is similar to the approach used in [23, 30, 46] where the authors noticed that the most common state $\mathbf{s}_0$, which is typically the silent state for neural data, is

likely to be well approximated. Thus, one may use

$$\hat{Z}_{\text{MostSampled}} \equiv \frac{\exp[-E(\mathbf{s}_0)]}{p_{\text{data}}(\mathbf{s}_0)} \tag{15}$$

as an estimator of $Z$. This is equivalent to approximating the right-hand side of Eq (13) by only including the term $\mathbf{s} = \mathbf{s}_0$ and replacing $p_{\text{pair}}(\mathbf{s}_0)$ with $p_{\text{data}}(\mathbf{s}_0)$. In principle, one can also consider the ratio $\exp[-E(\mathbf{s})]/p_{\text{data}}(\mathbf{s})$ for all sampled states and take the mean or median of the distribution of this quantity, producing other estimators $\hat{Z}_{\text{mean}}$ or $\hat{Z}_{\text{median}}$.

In Fig 4, we evaluate these estimators by comparing them to the true value of $Z$ for $N = 25$, where we have calculated $Z$ exactly by performing the summation over all states. The results in Fig 4A–4C clearly indicate that the estimator in Eq 14 outperforms the others. In Fig 4D and 4E, the resulting $\hat{G}$ is compared directly with $G$ which, again, was computed using exact summation. In S1 Appendix we further describe the properties of $\hat{Z}$, its relation to importance sampling, and show that its accuracy depends on the statistics of the energy gaps between states of the pairwise model. In addition, we perform tests on its performance for $N = 100$, showing (1) that it is close to the true $Z$ of simulated populations consisting of 20 independent groups of five

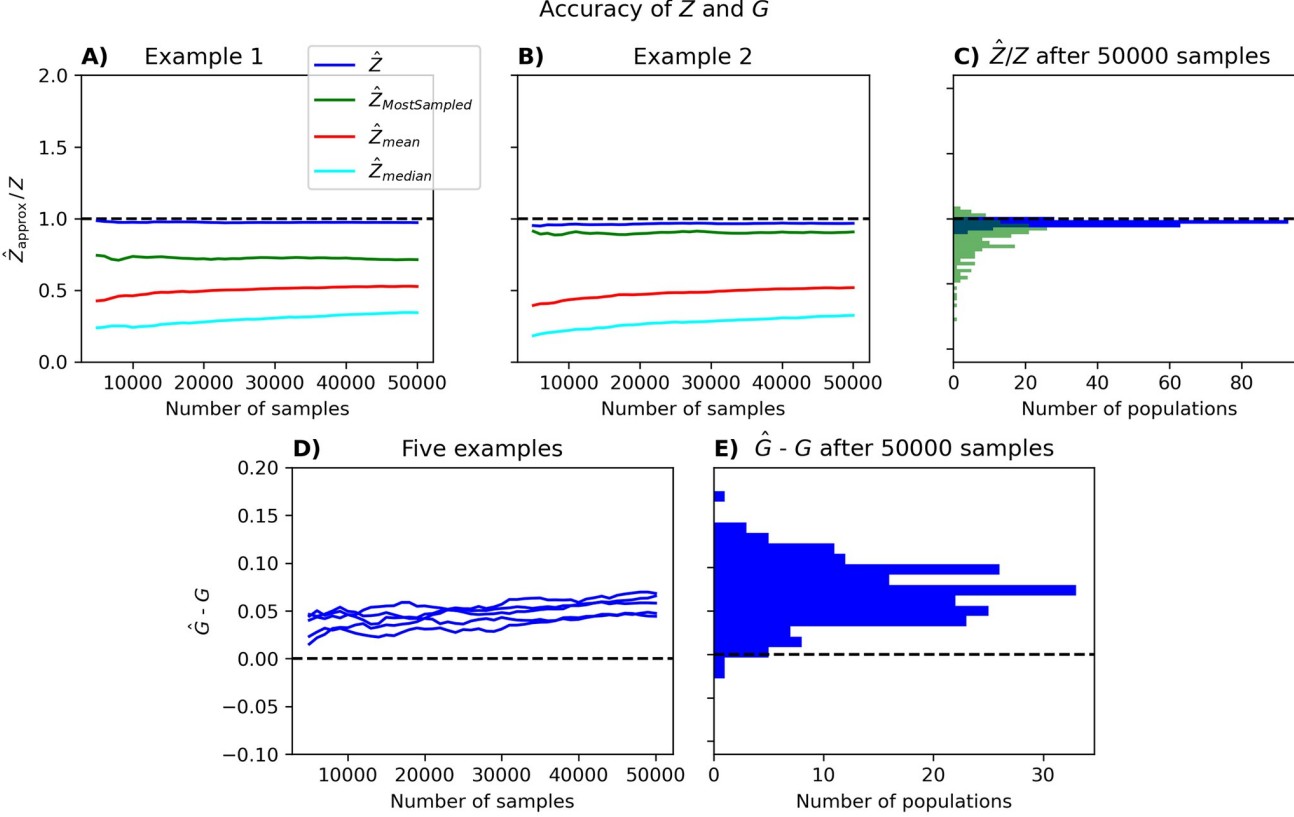

**Fig 4. Evaluation of our approximation of $Z$ and $G$.** A total of 50000 samples were randomly chosen from populations of 25 neurons. The four approximations of $Z$ were calculated using between 5000 and 50000 of these samples, in increments of 1000. Pseudolikelihood was used to estimate the parameters $h$ and $J$ that go into approximating $Z$. This procedure was performed for a total of 200 populations. Two examples are shown in panel A and B, where the ratio of the approximated $Z$ to the actual $Z$ is shown for $\hat{Z}$ (blue), $\hat{Z}_{\text{MostSampled}}$ (green), $\hat{Z}_{\text{mean}}$ (red), and $\hat{Z}_{\text{median}}$ (turquoise). (C) The ratio $\hat{Z}/Z$ (blue) and $\hat{Z}_{\text{MostSampled}}/Z$ (green) after 50000 samples for all 200 sets of Gaussian parameters. (D) Five examples of how $\hat{G} - G$ scales with the number of samples. (E) $\hat{G} - G$ after 50000 samples.

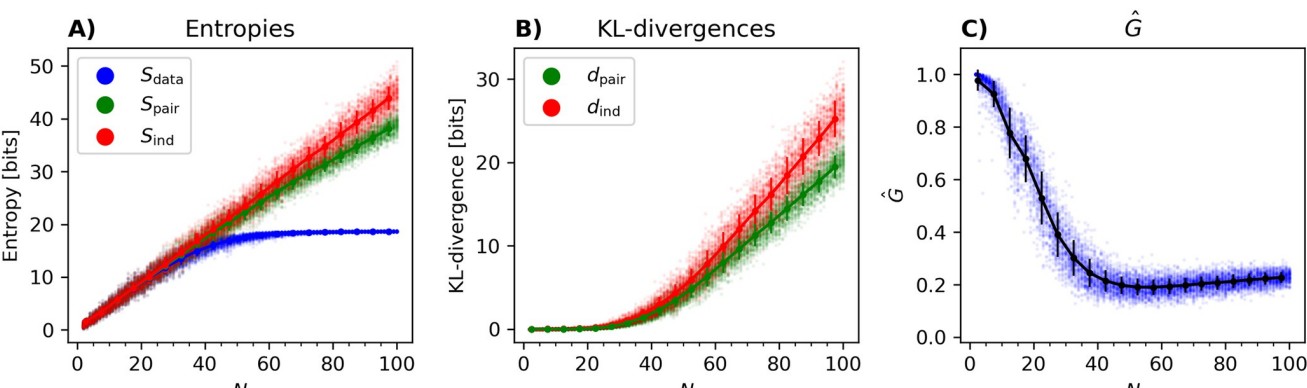

**Fig 5. Model properties versus *N* for large *N*.** Everything is the same as Fig 1, except that *N* is now extended to *N* = 100 using the estimator of $Z_{\text{pair}}$ explained in this section.

spins and (2) that it is close to other partition function estimators based on importance sampling and reverse importance sampling (e.g., [47]) for neural populations of *N* = 100 neurons. From this we see that $\hat{Z}$ is a good, reliable and efficient estimator of the partition function of the fitted PME, although we note that the resulting $\hat{G}$ slightly overestimates the true *G* (Fig 4).

We first use this way of estimating $Z_{\text{pair}}$ to extend the results in Fig 1 to population sizes of up to *N* = 100. The results are shown in Fig 5. As can be seen in Fig 5A, the entropy of the data saturates around *N* = 40, while the entropies of the PME and the independent model increase. The same applies to $d_{\text{int}}$ and $d_{\text{pair}}$ (Fig 5B). The decrease in *G* shown in Fig 1 can be seen to continue beyond *N* = 20 and reaches a shallow minimum for *N* ∼ 40 around *G* ∼ 0.2 and eventually plateaus (Fig 5C). It is possible that the performance would drop even more if $\hat{G}$ did not overestimate *G*. Note that $d_{\text{ind}}$ and $d_{\text{pair}}$ continue to increase with *N* beyond where *G* plateaus: in fact, the former quantities are so large (compared to the data entropy) that the small *G* reflects that the pairwise and independent models are both pretty bad models of the data. As noted above, this is different from the case where *G* is far from 1 because both models are equally good as was the case for *N* = 20 and small *δt* or $\bar{v}$ in Fig 2H and 2I.

We also plot the entropies, KL-divergences and the performance measure *G*, as functions of the perturbative parameter in Fig 6 for populations of up to *N* = 100. As expected from the perturbative expansion, we again see good performance for small $N\bar{v}\delta t$ followed by a drop as *N* increases. The inset in Fig 6C shows a zoom-in of the region *N* ≤ 20: similarity to Fig 2G shows that using our approximation of *Z* leads to results similar to the exact enumeration, at least when we can perform this enumeration.

In S3 Fig we show the same results as in Fig 6 but when the data are binned at *δt* = 0.01 *s* instead of *δt* = 0.02 *s*. As expected from the discussion in the previous section and from Fig 2I, this smaller time bin does not rescue the pairwise model for large populations. Furthermore, in S2 Appendix we show that our results are not substantially affected by the finite amount of data we have from the neural populations we have analysed here.

## Third-order correlations and probability of *m* active neurons

Instead of the performance measures based on KL-divergences, several studies have evaluated the performance of the PME model by comparing the third-order correlations in the data with

Performance of pairwise model up to N = 100

**Fig 6. Performance of the pairwise model inferred with pseudolikelihood from neural data, for large N.** 100 populations of size N were randomly chosen, where N varied from 2 to 100. For these populations, the entropy of the data $S_{\text{data}}$, the entropy of the PME model $S_{\text{pair}}$, and the entropy of the independent model $S_{\text{ind}}$ (A) were calculated. Then, the KL-divergences $d_{\text{pair}}$ and $d_{\text{ind}}$ (B) were used to calculate G (C). In panel C, the gradient represents the number of neurons in the population (blue [few] → turquoise [many]). For subpopulations with 15 or fewer neurons, G and $S_{\text{pair}}$ were calculated by summing over all states. For subpopulations with more than 15 neurons, $\hat{G}$ was calculated using $\hat{Z}$ from Eq (14). Lines represent means and standard deviations of G. This figure shows that $\hat{G}$ has an initial linear scaling with $N\bar{v}\delta t$ followed by a sharp fall and a plateau.

those predicted by the PME and the independent model [30, 31, 42, 43]. In addition, one can look at other statistical features of the data, such as the probability that $m$ neurons are simultaneously active in a time bin as in [23, 27, 30, 31, 42–44]. These alternative performance measures raise two questions: (a) What is the relationship between the various performance measures? (b) How does performance measured by these quantities change with population size?

In Fig 7, we show examples of third-order correlations, connected third-order correlations, and probability of $m$ simultaneously active neurons according to the data and as predicted by the PME and independent models. The results of a population with $N = 10$ neurons and a large G are shown in Fig 7A–7C. In this case, third-order correlations are well predicted, both by PME and independent models, although $G_C$ indicates that the pairwise model is better. The situation for $m$ simultaneously active neurons is similar. Connected third-order correlations are, however, very small and both the PME and independent model predict them very poorly. Fig 7D–7F show the same quantities with the same general conclusions, but now for a population with $N = 30$ where the performance according to $\hat{G}$ is very low. The same behavior is seen in Fig 7G–7I for a population of $N = 50$ neurons. These results indicate that the relationship between the different performance measures is not trivial. The model can do well in terms of predicting third-order correlations or the number of simultaneously active neurons, while performing well or poorly in terms of $\hat{G}$.

The nonlinear relationship between the various performance measures is depicted in Fig 8A–8C. In Fig 8D–8F, we plot $G_C$, $G_{\tilde{C}}$ and $G_H$ as a function of $N\bar{v}\delta t$ for 100 populations of $N = 5, 6, \ldots, 100$ random neurons. We see that in general $G_C$, $G_{\tilde{C}}$ and $G_H$ all drop as $N\bar{v}\delta t$ increases, though in the case of $G_{\tilde{C}}$ the drop is much smaller and even for small $N\bar{v}\delta t$ the PME is not much better than the independent one. Additionally, we notice that the standard deviations of $G_C$, $G_{\tilde{C}}$ and $G_H$ are much larger than that of $\hat{G}$, suggesting that the alternative performance measures may be less reliable. These results are shown without comparison to the independent model in S5 Fig.

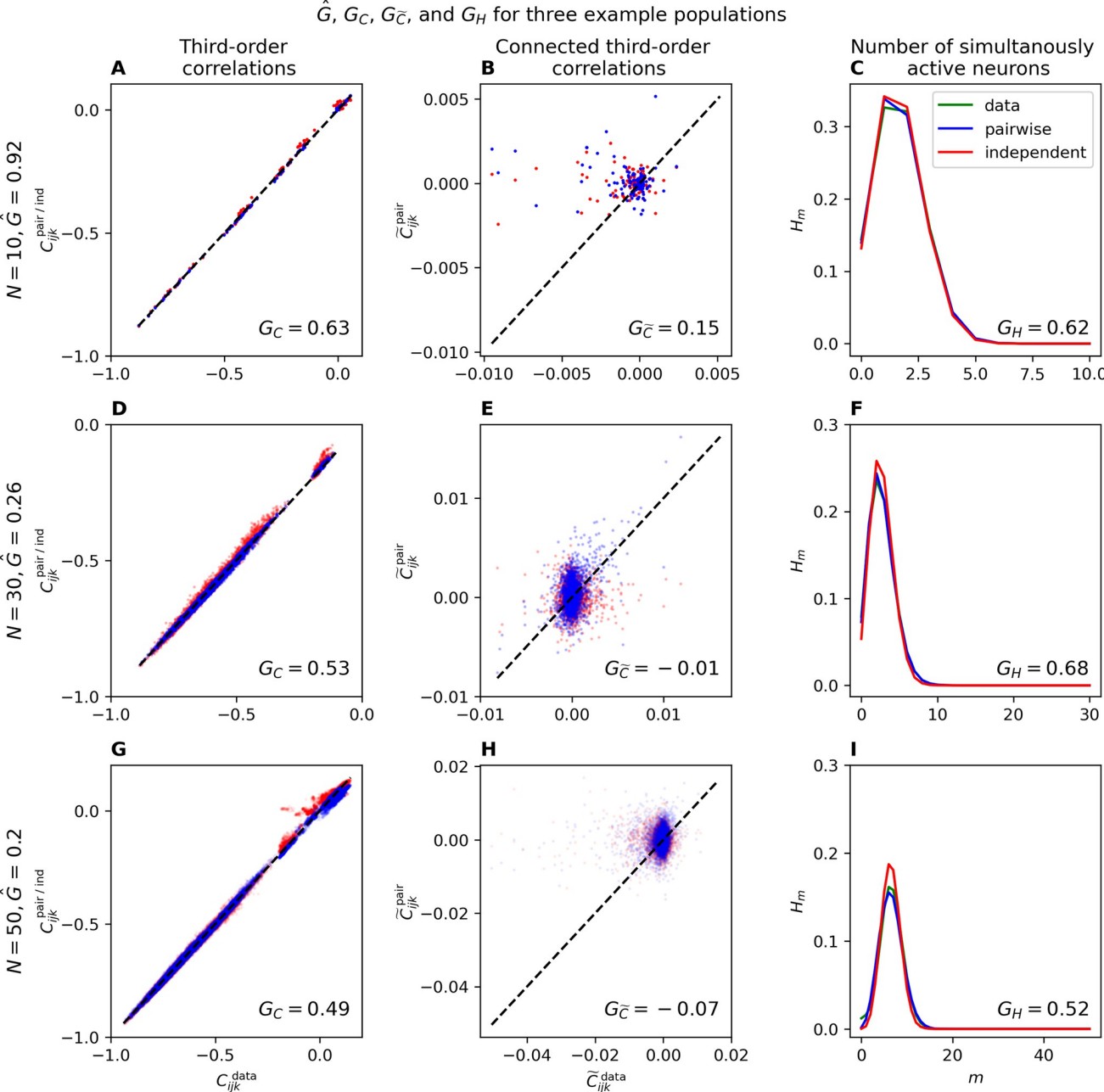

**Fig 7. Example populations for alternative performance measures.** The rows each contain an example population of $N = 10$ (A-C), $N = 30$ (D-F), and $N = 50$ (G-I) neurons. The columns show how well the inferred pairwise (blue) and independent (red) models predicts the third-order correlations, connected third-order correlations, and number of simultaneously active neurons observed in the data. $C_{ijk}^{\text{pair}}$, $\widetilde{C}_{ijk}^{\text{pair}}$, and $H_m^{\text{pair}}$ were calculated by sampling the pairwise model, using as many samples as in the data. This figure shows that the alternative performance measures $G_C$, $G_{\widetilde{C}}$ and $G_H$ often makes the pairwise model look better than $\hat{G}$ does.

## Performance on data from different cortical areas and experimental conditions

In the previous sections, we performed the analyses without taking into account the area from which the neurons are recorded. In this section, we analyze data from visual, auditory,

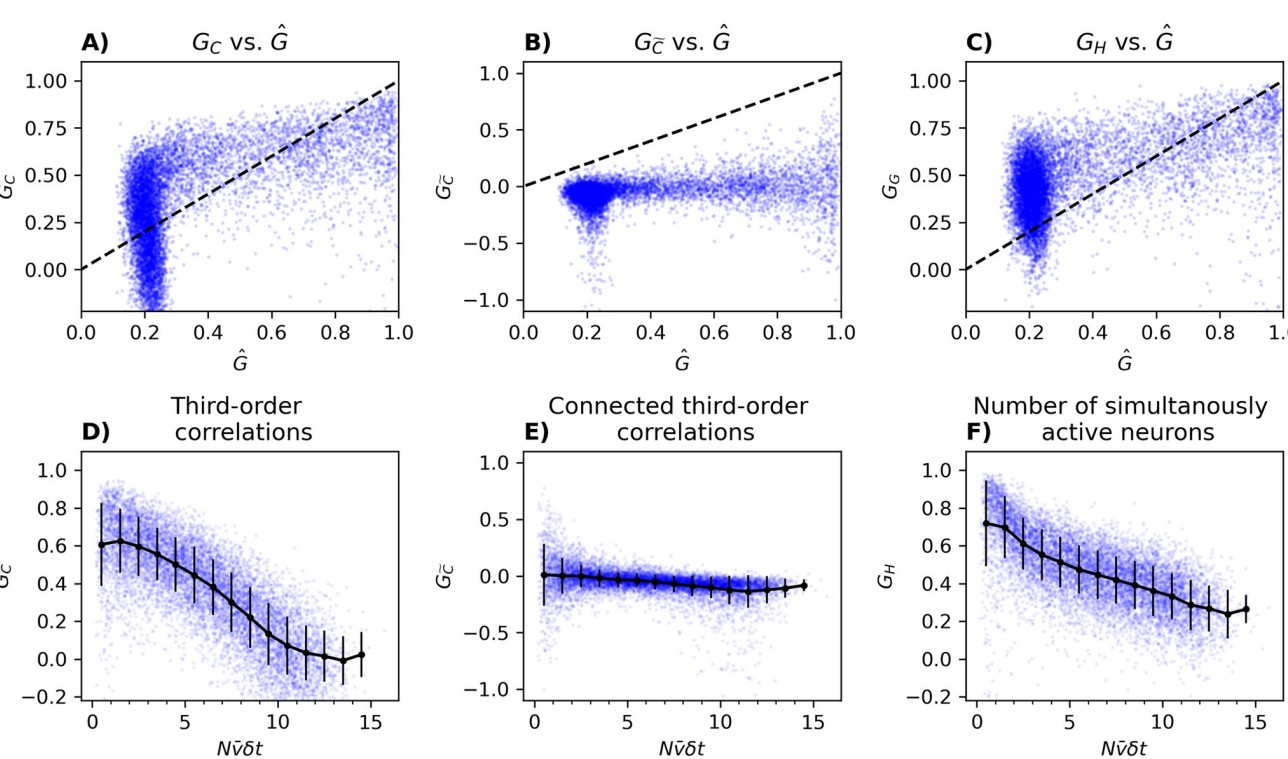

**Fig 8. Alternative performance measures versus $\hat{G}$ and as a function of $N\bar{v}\delta t$.** Panel A-C shows a scatter plot of $G_C$, $G_{\tilde{C}}$ and $G_H$ (from panel D-F) against $\hat{G}$ from Fig 6. Panel D-F shows how $G_C$ (D), $G_{\tilde{C}}$ (E), and $G_H$ (F) scales with $N\bar{v}\delta t$ for 100 randomly chosen populations of size $N$, where $N$ varied from 5 to 100. The black lines represent means and standard deviations of $G$. In panel D, E, and F, 408, 34, and 35 outliers with $G_C < -0.2$, $G_{\tilde{C}} < -1.1$, and $G_H < -0.2$ were omitted. This figure shows that $G_C$ and $G_H$, but not $G_{\tilde{C}}$, decreases a bit with $N\bar{v}\delta t$.

somatosensory, and motor cortices separately to see if there are qualitative or quantitative differences between the areas with regard to the performance of the pairwise model.

The results are shown in Fig 9, where we plot $\hat{G}$ versus $N\bar{v}\delta t$ for neurons from different areas of the cortical system separately. In all cases, we see a similar decline in $\hat{G}$ as a function of $N\bar{v}\delta t$.

In Fig 10, we consider the performance of the pairwise model for neurons recorded from the visual cortex during the periods of lights-on and lights-off conditions separately, for 5 populations of each size $N = 5, 10, 15, 20, 25, \ldots 100$ binned at $\delta t = 0.02$ seconds. Because we only have recordings of the same neurons for $\sim$ 20 minutes for each condition, we include $\hat{G}$s inferred from random samples (from all experimental conditions) constituting $\sim$ 20 minutes to control for the reduction in data length.

There are two features in Fig 10C that are important. First, we see that even though we only have $\sim \frac{1}{6}$th of the data we had in Figs 2 and 6, $\hat{G}$ exhibits a behavior similar to those cases. Second, we note that the decay in the performance of the PME model in the dark is faster and becomes worse compared to the light condition, which behaves as random samples.

To better understand this, we plot $d_{\text{ind}}$ and $d_{\text{pair}}$ for the same neurons in lights-on versus lights-off conditions in Fig 10A and 10B, color coded by population size. We see that, in general, the pairwise model is slightly better (smaller $d_{\text{pair}}$) under the lights-on condition

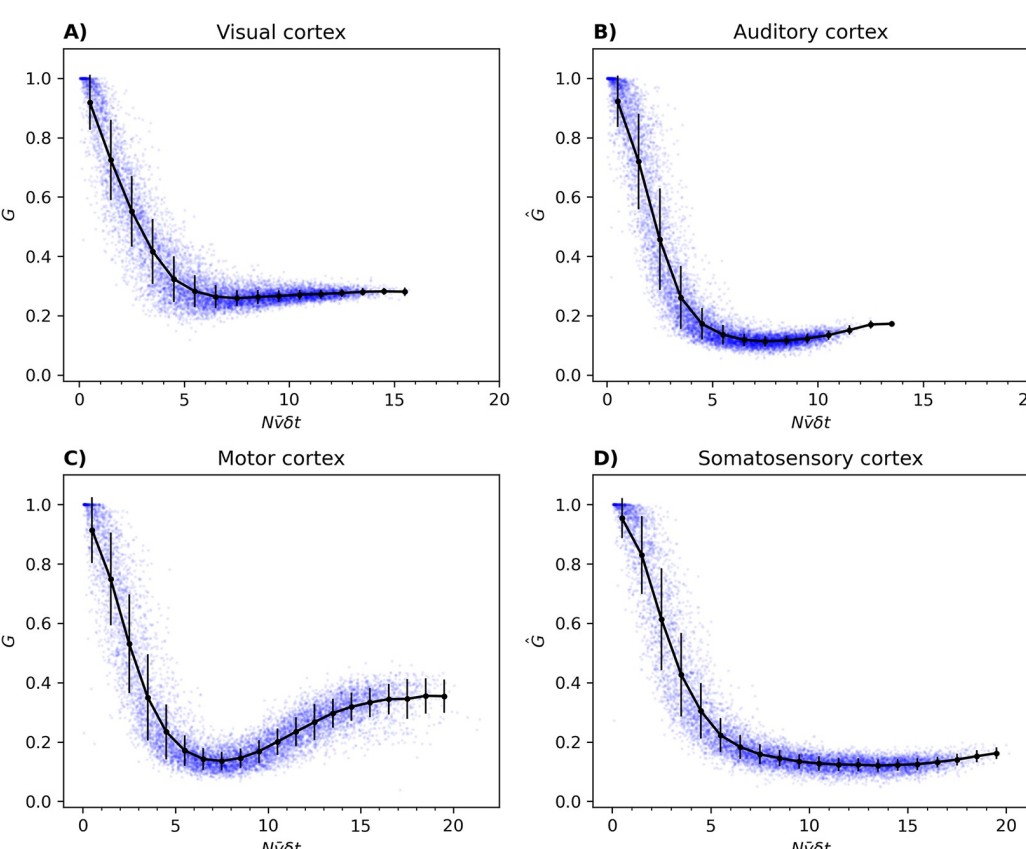

**Fig 9. Performance of the pairwise model inferred with pseudolikelihood for populations from (A) the visual, (B) auditory, (C) motor, and (D) somatosensory cortices.** For each $N$, 100 populations were randomly chosen from 539 neurons in the visual cortices, 376 neurons in the auditory cortex, 1115 neurons in the motor cortex, and 287 neurons in the somatosensory cortex. A bin size of $\delta t = 0.02$ was used for the visual and auditory cortices, while for the motor cortex and somatosensory cortices the bins were 0.06 seconds and 0.14 seconds, respectively. These reflected differences in the mean firing rate $\bar{v}$ of populations from different cortical areas with a mean of $M = 6.27$ and a standard deviation of $SD = 1.24$ in the case of the visual cortex, $M = 4.98$ and $SD = 1.11$ for the auditory cortex, $M = 2.67$ and $SD = 0.64$ for the motor cortex, and $M = 1.17$ and $SD = 0.24$ for the somatosensory cortex.

compared to the lights-off condition. On the other hand, the independent model is more clearly worse in the lights-on condition (larger $d_{\text{ind}}$) than in the lights-off condition.

In Fig 11A–11D, we show a similar analysis for neurons recorded from the auditory cortex during the periods of sound-on (bursts of white-noise) and sound-off (silence). We see no substantial difference between the two conditions. This may have been a result of the sound-on condition only amounting to ∼6 minutes of data. This is also the reason we only go up to $N = 15$ in this case. To confirm that the lack of a difference is not just due to limited data, we downsampled the light-on and light-off conditions to the same data length, and still find worse performance of the PME during darkness (Fig 11E–11H).

## Couplings within and between cortical areas

Another way to investigate what the PME model captures about neural function is to look at its inferred couplings $J_{ij}$. First, we might suspect that neurons from the same cortical area are

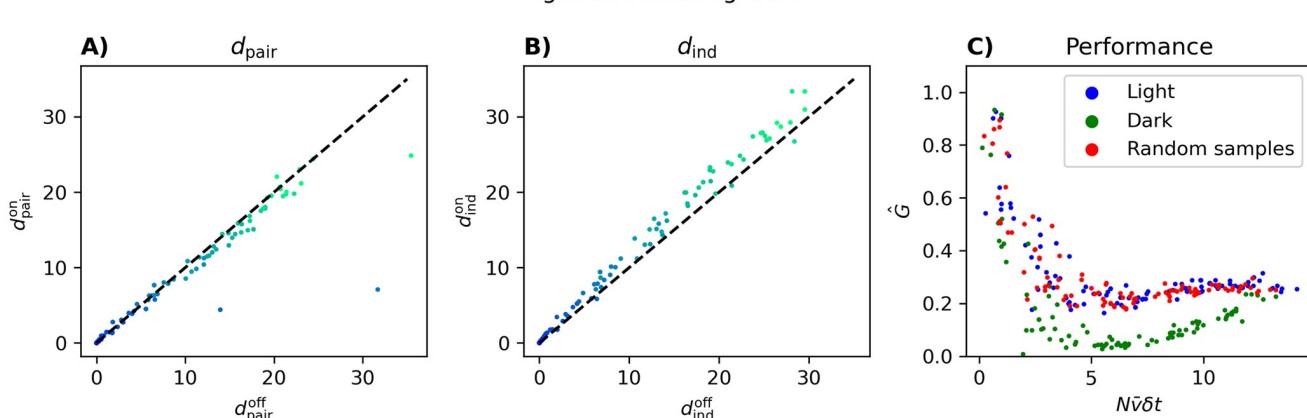

**Fig 10. Performance of the pairwise model inferred from data recorded while foraging with the light on and with the light off.** Five populations of size $N = 5, 10, 15, 20, 25, \dots 100$ were chosen randomly. Then, $d_{\text{pair}}$ (A), $d_{\text{ind}}$ (B), and $\hat{G}$ (C) was calculated only from samples obtained with the light on (blue), from samples obtained with the light off (green), or from $\sim 20$ minutes worth of random samples (red). The light being off results in 7 outliers with $\hat{G} < 0$, which were omitted. Pseudolikelihood was used to infer the parameters $h$ and $J$ for each of the three sets of samples. This figure shows that segregating the data into different experimental conditions and having only $\sim \frac{1}{6}$th of the data available in Fig 6, does not change the values of $\hat{G}$ substantially.

more related to each other than neurons from different cortical areas. This should be reflected in the magnitude of the $J_{ij}$s. To test this, we use the couplings inferred for $N = 100$ in Figs 5 and 6 and show that the mean absolute value of the $J_{ij}$s is larger when neuron $i$ and neuron $j$ is from the same area (Fig 12).

Second, if these couplings reflect a genuine relationship between two neurons, one might hope that their order is stable in the presence of other neurons. We test this in Fig 13, where we follow the five largest and five smallest couplings initially inferred in a random population of 20 neurons, as they are inferred in larger populations. The initial population could be from either visual or auditory cortices, and the neurons added to this initial population could also be from either visual or auditory cortices. In general, we see that the ordering of the strongest couplings is well-preserved in different "contexts". Additionally, the strongest couplings change more if the neurons being added are from the same area as the initial population.

## Performance when using approximate parameters

In the previous sections, we inferred the couplings of the pairwise model using the pseudo-likelihood approach, given its established high accuracy [15, 19, 40, 41]. This method, however, is one amongst a plethora of approximate methods for inferring the couplings that bypass the slowness of exact Boltzmann learning. These methods mainly depend on results from the analysis of the SK model [20] (see also the next section). Although comparisons of these methods with exact Boltzmann learning have been performed [9], the quality of the PME model they identify, as a statistical model, has not been studied. In particular, it is unclear how the inaccuracies of these methods in inferring the couplings affect the performance of the fitted PME model. This is what we address in this section for neural data.

Using a subset of the populations analyzed in Fig 6, we first show in Fig 14 that our results regarding how $G$ behaves as a function of $N$ do not depend on whether we use Boltzmann learning or pseudo-likelihood. In Fig 15, we show the $N\bar{v}\delta t$ dependence of $G$ when the pairwise model is inferred using the naive Mean-Field (nMF), Thouless-Anderson-Palmer (TAP),

Entropies differences for different stimuli

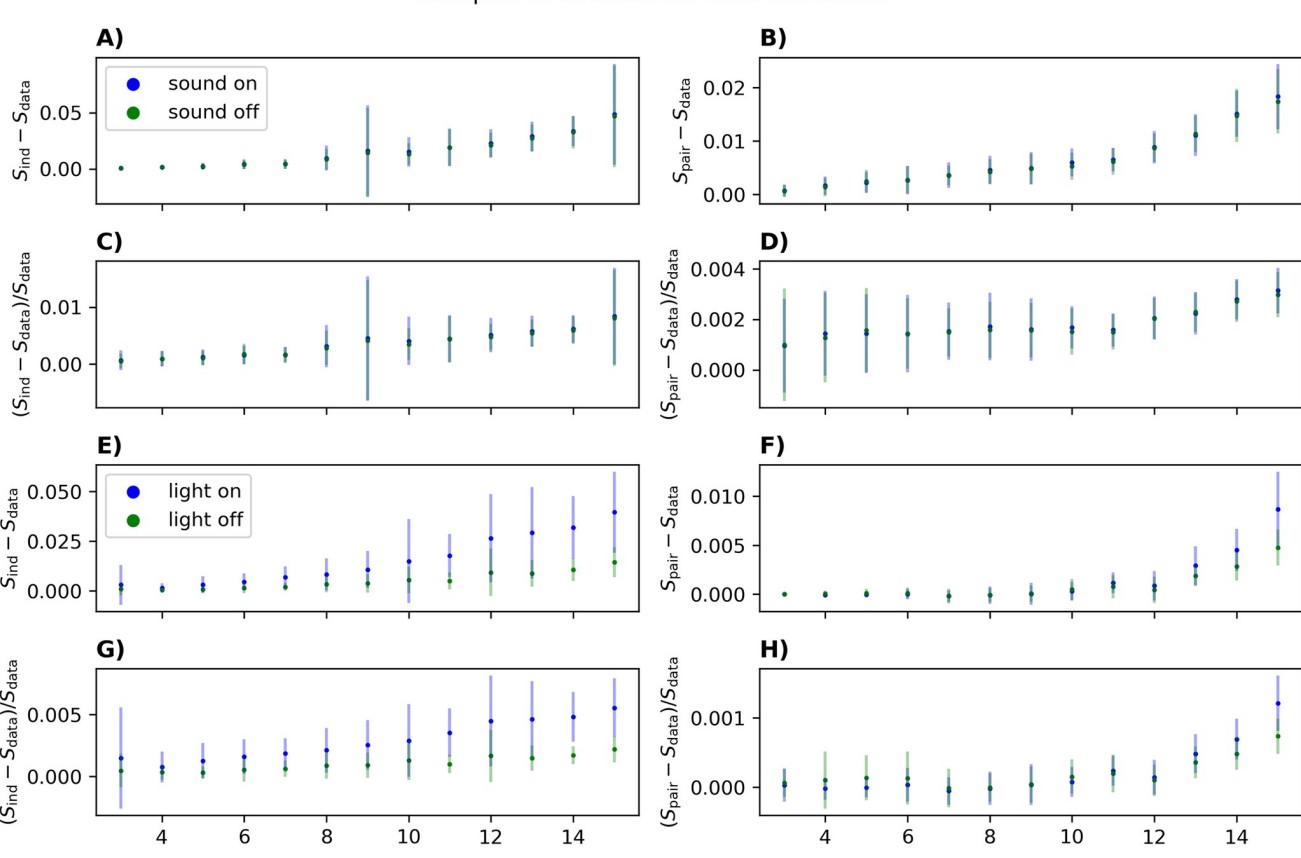

**Fig 11. Performance of the pairwise model inferred from data recorded under different sensory conditions.** Panel A-D shows data recorded from the auditory cortex, split into time points of silence and time points with a white-noise playing. Panel E-H shows data recorded from the visual cortex, split into periods with the light on and light off. For both areas, 50 populations were randomly picked for each $N$ between 3 and 15. The error bars represent the standard deviation of these populations. All data sequences were downsampled randomly to match the condition with the shortest duration (sound on). All parameters were inferred with Boltzmann learning with a learning rate of $\eta = 0.01$ and 50000 iterations (calculating $\langle s_i s_j \rangle_{\mathrm{pair}}$ and $\langle s_i \rangle_{\mathrm{pair}}$ exactly).

Independent-Pair (IP) and Sessak-Monasson (SM) approximation; see [19] for a review of the details of these methods. Compared to the results in Figs 6 and 14, we can draw the same general conclusions: as $N\bar{\nu}\delta t$ increases, the performance decays rather rapidly. We can also see that in the narrow range where $G \sim 1$, TAP, SM, and IP achieve slightly higher mean performance than nMF, but that the performance of IP decays to lower values, showing few signs of saturation as $N\bar{\nu}\delta t$ increases. This is perhaps not surprising, given that in IP, the coupling $J_{ij}$ between pairs of neurons is found by ignoring all other neurons in the population.

## Complexity of the pairwise correlations

In this section, we turn our attention to the issue of the complexity of the inferred model. Complexity quantifies how rugged the free energy is, which, in turn, is reflected in the number of metastable states of the model, $N_{\mathrm{ms}}$. The number of metastable states of the pairwise model fitted to retinal data has been investigated in [30], concluding that this number increases exponentially with $N$. Groups of similar metastable states were shown to be active in response to repetitions of the same stimulus. The results were therefore interpreted as evidence of powerful error correction mechanisms, with metastable states acting like memories stored in a Hopfield

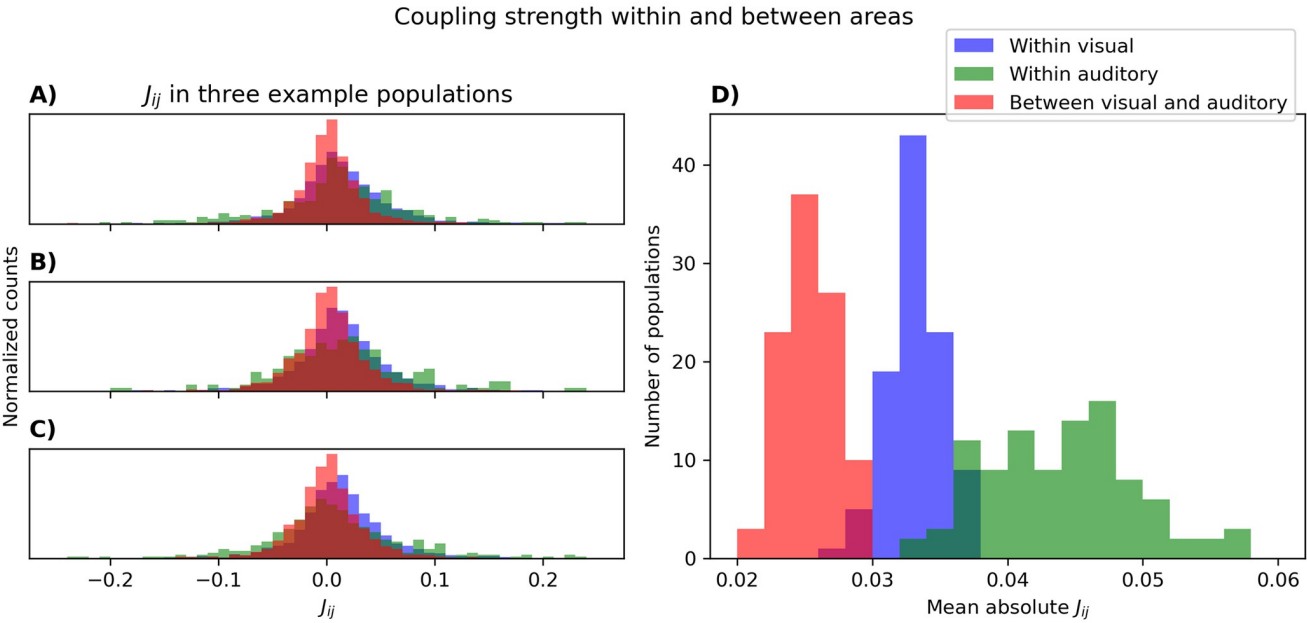

**Fig 12. Strength of the couplings $J_{ij}$ within and between visual and auditory cortices.** Here we use the $J$s inferred for $N = 100$ in Figs 5 and 6. Panel A-C display the couplings between visual cortex neurons (blue), between auditory cortex neurons (green), and between visual and auditory cortex neurons (red) for three of the 100 populations. In panel D we display the mean absolute $J_{ij}$ between visual cortex neurons (blue), between auditory cortex neurons (green), and between visual and auditory cortex neurons (red) for all 100 populations.

network. However, in that study, metastable states were defined as configurations stable with respect to one spin flip. Since stability with respect to one spin flip is unlikely to be a good indicator of stability in general, in this section we use a more accurate definition of complexity and metastable states following the classical work on the SK model [33, 48].

The SK model is simply a distribution identical to Eq (2), where the couplings $J_{ij}$ are assumed to have been drawn from a Gaussian distribution with mean $J_0/N$ and standard deviation $J_1/\sqrt{N}$, where $J_0$ and $J_1$ do not depend on $N$ and the limit $N \to \infty$ is taken. Metastable states are identified with the solutions $m_i$ of the TAP equations [49], given the couplings $J_{ij}$ and fields $h_i$ [33],

$$m_i = \tanh[h_i + \sum_j J_{ij}m_j - \sum_j J_{ij}^2(1 - m_j^2)], \tag{16}$$

which are the minima of the TAP free energy (see also S3 Appendix). The complexity is then defined as $\Sigma = N^{-1} \log N_{\text{ms}}$ where $N_{\text{ms}}$ is the number of such solutions. In a nutshell, the complex phase arises when the system exhibits many frustrated configurations: situations where, for example, for a triplet of variables, $s_i$ and $s_j$ prefer to have the same sign as $J_{ij} > 0$, $s_i$ and $s_k$ prefer to have the same sign as $J_{ik} > 0$, but $s_j$ and $s_k$ prefer to have opposite signs as $J_{ik} < 0$. These scenarios are more likely to occur when $f \equiv \text{std}(J_{ij})/\text{mean}(J_{ij})$ is large [21]. When $f \to 0$, on the contrary, we have the normal phase. A careful analysis of the SK model shows that $\Sigma = 0$ in the normal phase, but that when the spin-glass susceptibility (see S3 Appendix) diverges for $N \to \infty$, $\Sigma > 0$, that is, $N_{ms}$ grows exponentially with $N$. Furthermore, this can be shown to happen when $J_1^2 S = 1$ with $S \equiv N^{-1}\sum_i(1 - m_i^2)^2$: the normal phase becomes unstable when $J_1^2 S > 1$; see also S3 Appendix for more details.

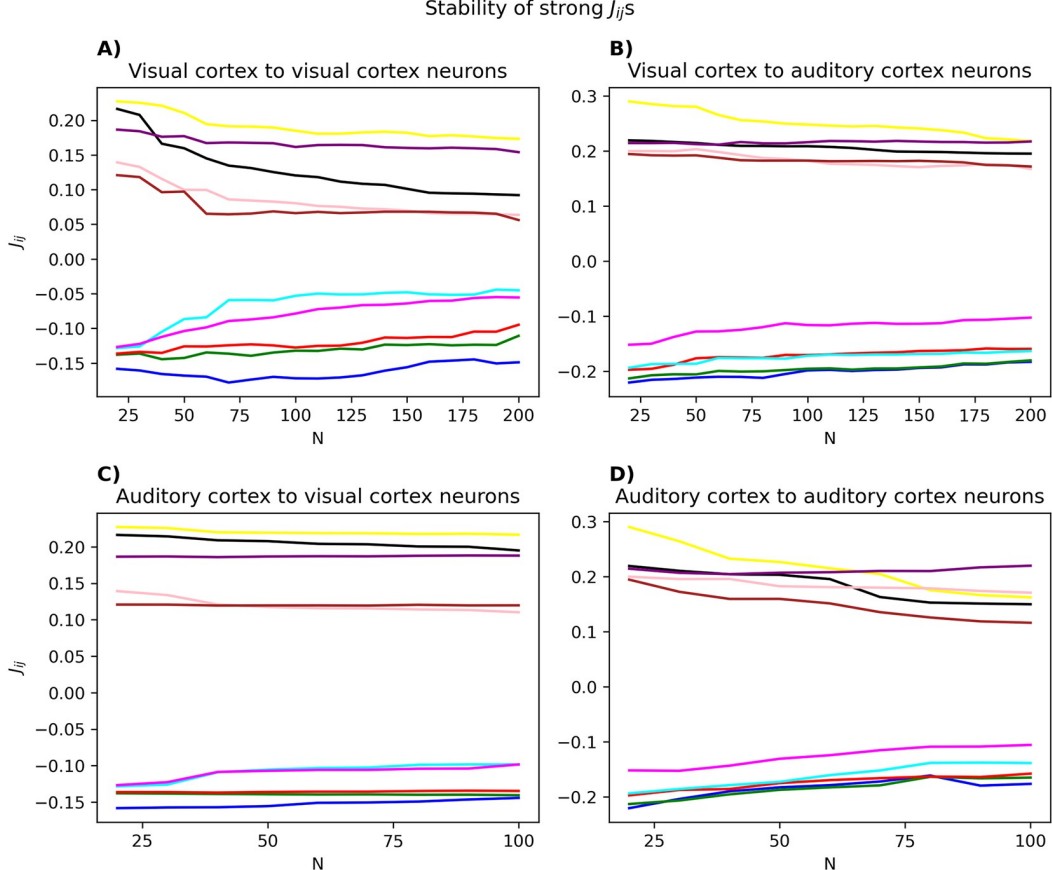

**Fig 13. Stability of strong couplings inferred from a population of 20 neurons, as more neurons are being added.** The five largest and five smallest couplings after inference on 20 random visual (A and C) or auditory (B and D) neurons were picked. The value of these couplings were then tracked when they were inferred again in larger populations of visual (A and B) or auditory (D and C) neurons. All parameter inference was done with pseudolikelihood maximization.

In the case of a PME model fitted to neural data, following [9], in Fig 16A and 16B, we first plot the $N$ dependence of the mean and standard deviation of the couplings inferred from neural data. We can also calculate, for each value of $N$, given the SK assumptions, what the mean and standard deviation of the couplings would be, given the means and correlations of the spin variables from the data [9]; see S3 Appendix.

The first important observation is that the mean couplings are generally very close to zero and get even closer to zero as $N$ increases. This is true for both the approximate inference and the SK prediction, although, for example, TAP leads to slightly smaller means. Second, the standard deviation of the couplings is much larger than their mean. Fig 16C shows that, in fact, $f$ increases with $N$ for the fitted models. According to the simple argument above, this implies that for large population sizes, there is a more significant chance of having frustrated configuration. The fact that more and more such potentially frustrated subsets of neurons appear in the data is illustrated by the fact that the spin-glass susceptibility (see definition in S3 Appendix) increases with $N$ (inset in Fig 16C).

In general, the $N$ dependence of the mean and standard deviation of the inferred couplings and the SK model are similar, although not perfect. Consistent with the analysis of the simulated model [9], the important features, i.e. the decay with $N$, and the standard deviation

**Fig 14. Performance of the pairwise model inferred with Boltzmann learning.** Ten populations of size $N$ = 20, 40, 60, 80, 100 were chosen randomly from Fig 6. For these populations, the parameters $h$ and $J$ were inferred using Boltzmann learning with 50000 iterations, 25000 samples per iteration, and a learning rate of $\eta$ = 0.001, starting from the pseudolikelihood approximation. Here, we see the $\hat{G}$s resulting from this (blue), in addition to the $\hat{G}$s resulting from pseudolikelihood (green; as in Fig 6). This figure shows that using pseudolikelihood instead of Boltzmann learning does not change our conclusions.

getting comparatively larger than the mean, are there. Turning to the stability condition of the normal phase, $J^2S < 1$, in Fig 16, we plot $J_1^2S$, where $J_1$ is calculated using either approximate inference methods or Eqs. (4) and (9) in S3 Appendix. In both cases, $J_1^2S$ increases with $N$, but does not reach the critical value 1. If this trend continued, a linear extrapolation from the range $N = 200 - 400$ predicts that the critical value is reached when $N \sim 2500$ if we use the results in S3 Appendix and $N \sim 6000$ if TAP or pseudolikelihood is used.

Of course, none of this means that there is a transition or increased metastability in the brain: these statements are instead about the PME model fitted to the data. To the extent that this relates to the brain, it shows that as the population size increases, the large-scale structure of pairwise correlations becomes more complex. This means that there are many positive and negative correlations, which may be individually weak, but strong enough such that the PME model (which only fit pairwise correlations) should develop more metastable states as $N$ increases to fit such correlations. In other words, because many metastable states come from many frustrated couplings $J_{ij}$, one may suspect that these couplings were fitted on many conflicting pairwise correlations between neurons.

## Discussion

The pairwise maximum entropy model is a popular model for studying complex systems ranging from the immune system [50] and proteins [51] to neuronal networks [6, 7, 10]. The PME model studied here in some sense plays the role of the Gaussian distribution for binary variables: it is the maximum entropy model given only the mean and standard deviations of its variables. Consequently, it is a natural choice for a first attempt at modelling the joint distribution of spikes in neural populations. In this paper, we systematically evaluated the performance of the PME model for large population sizes, different bin sizes, different population firing

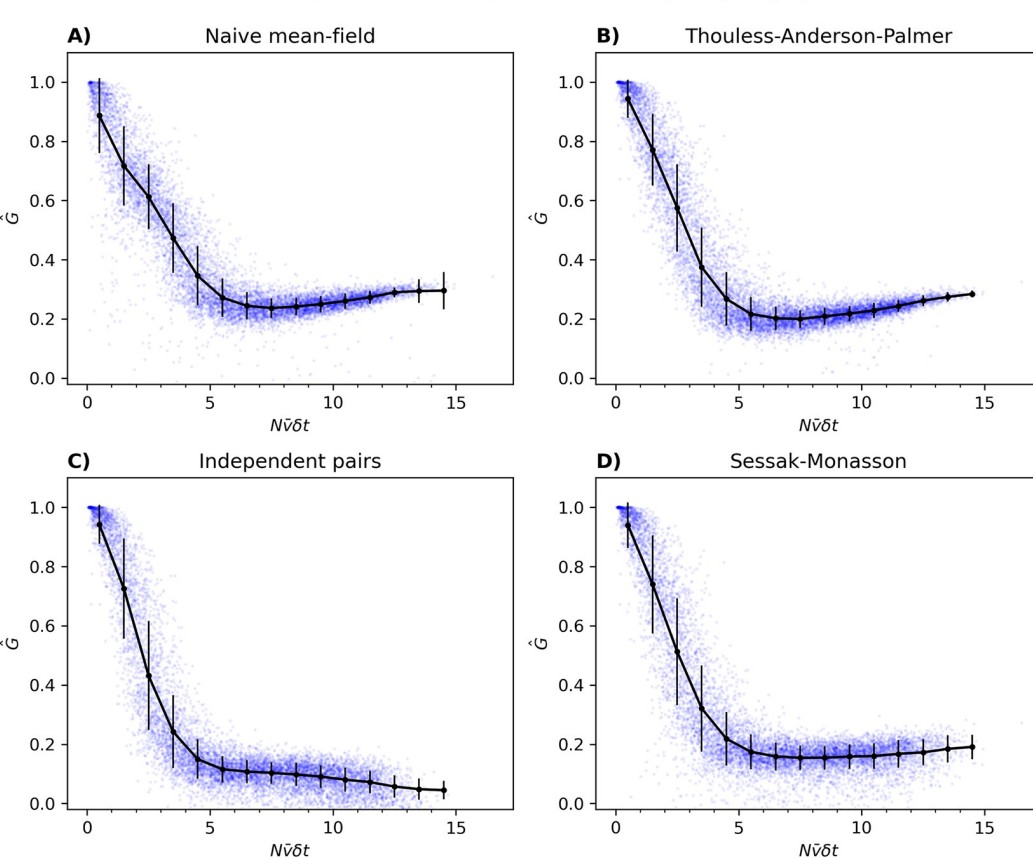

**Fig 15. Performance of the pairwise model inferred using nMF, TAP, IP, or SM from neural data, for large N.** (A-D) Identical to Fig 6, except that nMF, TAP, IP, and SM have been used to approximate the pairwise model. Using nMF, TAP, IP, and SM resulted in 672, 60, 655, and 288 outliers with $\hat{G} < 0$, respectively, which were omitted. Additionally, for the SM approximation, 385 (out of 9800) $\hat{G}$s were completely omitted due to overflow errors. This figure shows that inaccurate parameters do have an effect on $\hat{G}$, but the characteristic scaling persists.

rates, and different cortical areas. We also performed this evaluation according to different measures of performance. Like many previous studies [6, 7, 23–27], we find that the pairwise model exhibits excellent performance for small $N$.

Despite the excellent performance for small populations, and consistent with previous theoretical predictions [10, 46, 52, 53], our analysis shows that the excellent performance of the PME model observed for small populations does not extend to larger populations. This is most clearly reflected in the quantities $G$ and $\hat{G}$. For small populations $N < 10$, the KL-divergence between the data distribution and the pairwise model $d_{\text{pair}}$, is only a few percent of that of the independent model $d_{\text{ind}}$, which is itself quite small, leading to values of $G$ close to one. This ratio is on average $\sim 20\%$ ($G \sim 0.8$) for $N = 10 - 15$, but saturates to 80% (that is $\hat{G} \sim 0.2$) for $N \sim 40$. Although this performance can generally be improved some by considering smaller population firing rates and/or smaller bin sizes, this improvement does not take the PME model to the excellent performance regime of small population sizes. These results are quite generic: they are independent of whether the neurons in the population were a mix of neurons from auditory or visual cortices, if data only from single areas were used, or if data from visual

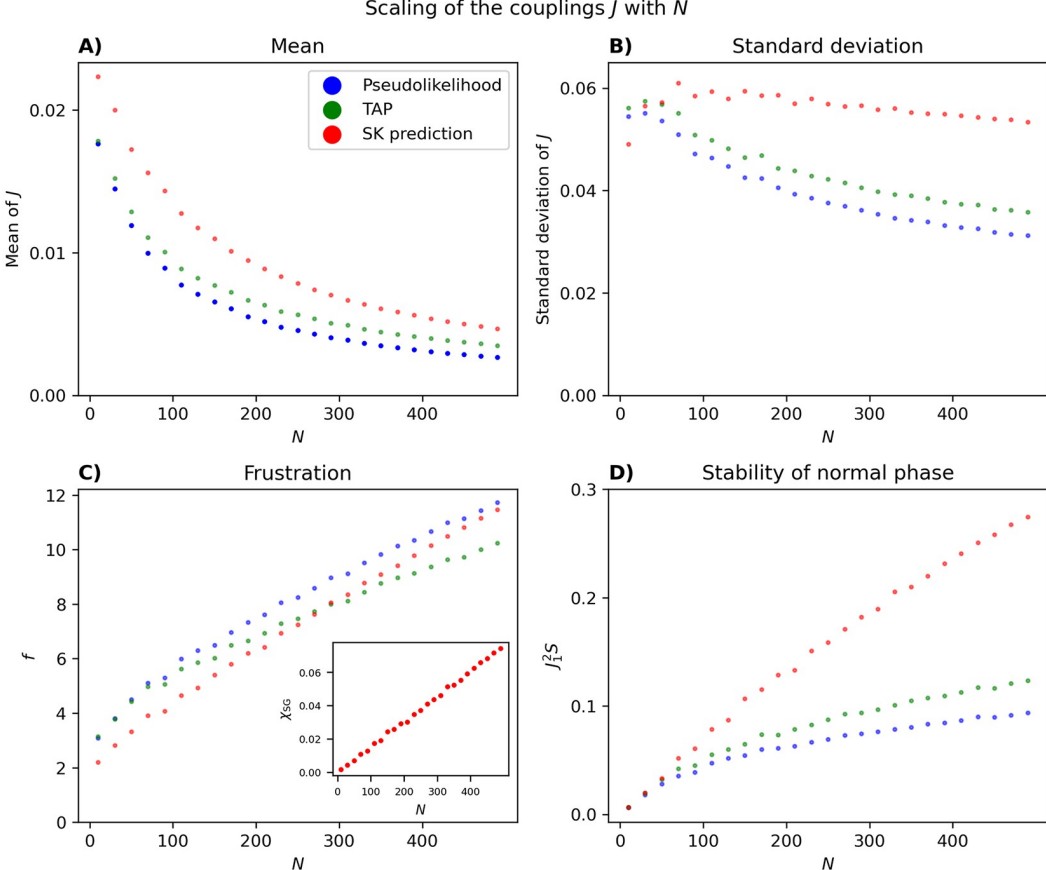

**Fig 16. The scaling with _N_ of the couplings and entrance into the spin-glass phase.** For each _N_ between 10 and 490, in increments of 20, an average was taken over 50 randomly chosen populations. Mean (A) standard deviation (B) of the couplings as inferred by pseudolikelihood (blue), TAP (green), and predicted from Eqs. (4) and (9) in S3 Appendix (red). (C) The ratio of the standard deviation of the the couplings to their mean. Inset, the spin-glass susceptibility versus _N_. (D) The quantity $J_1^2 S$.

and auditory cortex were analyzed separately for different sensory conditions. The results were also independent of whether we used exact Boltzmann learning or any of Pseudo-Likelihood, naive Mean-Field, TAP, IP, or SM approximations to infer the parameters of the model.

In addition to measuring performance using KL-divergences we also, like many others [8, 23, 27, 30, 31, 42–44, 54, 55], used third-order correlations, connected third-order correlation, and the number of simultaneously active neurons. We find that while $G_C$, $G_{\tilde{C}}$, and $G_H$ also fall as _N_ increases, they decrease far slower than $\hat{G}$. Thus, the sharp drop of $\hat{G}$ suggests that the PME model does not perfectly capture the entire correlation structure of the data. This further hints that higher-order correlations might be important for understanding the statistics of neural firing. However, from the slower drop in $G_C$, $G_{\tilde{C}}$, and $G_H$, it seems that a wide collection of higher-order correlations can be informative.

Finally, we evaluated the scaling of the couplings of the PME model as _N_ increases, using the approximate inference methods and the expression for the mean and standard deviation of the couplings that can be derived for the mean-field SK model. We found that, in all cases, both the means and standard deviations of the couplings decrease with _N_ in a manner similar to the SK model, and that the standard deviation is much larger than the mean. We saw that,

as a simple measure of the complexity of the model, the ratio between the standard deviation of the couplings and their mean increases with $N$, signaling an increasing degree of potentially frustrated groups of cells. We then followed the standard definition of metastable states based on the solution of the TAP equations [9, 33]. This allowed us to use results from the spin-glass literature. Despite the likely increase in frustrated states, even the largest population is technically outside the complex phase. However, we observed that as $N$ increases, the model approaches the complex phase. Perhaps due to the more stringent definition of metastable states used here, this is inconsistent with [30], who found an exponential growth of metastable states in the retinal data even for much smaller populations. However, the general message that the free energy of the model becomes more rugged as $N$ increases is still true because, as $N$ increases, the model gets closer to the phase where $N_{ms} \sim \exp(N)$, and could indeed enter this phase at larger populations of $N \sim 2500 - 6000$. One possible interpretation of this result is that the large number of metastable states act as attractors performing error correction. However, this interpretation makes the assumption that the properties of the metastable states of the pairwise fit are close to those of the true distribution. The fact that for large $N$, the pairwise model has a large KL-divergence from the true distribution makes this unlikely. Another interpretation, consistent with the large $d_{pair}$, is that the increasingly rugged free energy reflects the difficulty that the pairwise model has in fitting the data: as $N$ increases it can only fit it by developing many metastable states, as it has to fit an increasing number of positive and negative correlations whose amplitudes do not decrease with $N$. The increased ruggedness, in turn, makes sampling from the model, and thus computing with it, more difficult. Perhaps incorporating higher-order correlations [28, 56], time-varying and common external input [57, 58], dynamics [9, 59–61], or the effect of hidden neurons [62, 63] leads to models that can fit the data without entering into a phase where sampling is difficult.

We note that our results do not mean that the PME model is not useful. Depending on what the model will be used for, even small improvements over the independent model by adding pairwise correlations for large $N$ could still be very important and quite informative [64, 65]. It is just that the model is not an excellent match to the data and does not universally capture the probability of different activity patterns, as it does for small populations. Therefore, general conclusions from these models may be dangerous. In fact, the usefulness of the PME model for different purposes is reflected in its performance according to other performance measures $G_C$ and $G_H$. Because these measures retain higher performance as $N$ increases, the PME model could be a useful tool if, say, predicting third-order correlations is of importance in a given application. As we found here in visual cortex, the performance of pairwise and higher-order models can also vary depending on external inputs in differing experimental conditions. In the present case, we showed that the PME model was useful for modeling spiking interactions among smaller numbers of visual cortical neurons with the lights on, which might reflect preferential functional connectivity among similarly tuned neurons [66]. With the lights off, however, this tuning might drive the neural activity less, resulting in a better independent model. In any case, comparing lights-off with lights-on and sound-off with sound-on highlights that the PME model may capture the spiking statistics of the same network differently under different sensory regimes. Another potentially useful feature of the PME model is the assignment of relatively stable and informative functional relationships, defined by the inferred $J_{ij}$s, that takes into account at least some level of network effects, in contrast to pairwise correlations.

In summary, in this paper, we made a comprehensive quantitative study of the performance of the PME model as a probabilistic model for neural populations. We also discussed a number of ways that the PME model can be used as a conceptual tool for understanding neural coding, despite its failure for large populations. In fact, one can take the failure of the pairwise model as an indicator of the importance of higher-order correlations. In this way, the PME model

can be used to quantitatively study the role of higher-order correlations through the cortical hierarchy or layers of artificial neural networks [35].

## Supporting information

**S1 Fig. Effect of the bin size and maximum population size on the results in Fig 2** **Everything is the same as in Fig 2.** Everything is the same as in Fig 2, except that $\delta t = 0.01$ in the first column and $N = 10$ in the second and third column.
(PNG)

**S2 Fig. Entropies, KL divergences and *G* versus *N*.** The first column in S1 Fig, except that $N\bar{\nu}\delta t$ on the x-axis is replaced by $N$.
(PNG)

**S3 Fig. Effect of bin size Fig 6.** Everything is the same as in Fig 6, except for $\delta t = 0.01$.
(PNG)

**S4 Fig. Effect of lights-on and off on the KL-divergences.** Everything is the same as in Fig 6, except for $\delta t = 0.01$.
(PNG)

**S5 Fig. Predicting third-order correlations, connected third-order correlations, and the number of simultaneously active neurons.** Root mean squared error of $C_{ijk}$ (A), $\widetilde{C}_{ijk}$ (B), and $H_m$ (C) between the data and the pairwise model (blue) and between the data and the independent model (green). This is an alternative visualization of the data presented in Fig 8.
(PNG)

**S6 Fig. Boltzmann learning against pseudolikelihood maximization for neural data.** (A-B) A random subpopulation of 20 out of the 495 neurons were chosen. The pseudolikelihood parameters $h_i^{\mathrm{PL}}$ and $J_{ij}^{\mathrm{PL}}$ were plotted against the Boltzmann learning parameters $h_i^{\mathrm{boltz}}$ and $J_{ij}^{\mathrm{boltz}}$, which used a learning rate of $\eta = 0.01$, 40000 iterations, and 10000 samples per iteration. (C-D) A random subpopulation of 100 out of the 495 neurons were chosen. The pseudolikelihood parameters $h_i^{\mathrm{PL}}$ and $J_{ij}^{\mathrm{PL}}$ were plotted against the Boltzmann learning parameters $h_i^{\mathrm{boltz}}$ and $J_{ij}^{\mathrm{boltz}}$, which used a learning rate of $\eta = 0.01$, 80000 iterations, and 50000 samples per iteration. Further comparisons are available in [67].
(PNG)

**S1 Appendix. Properties and further tests of $\hat{Z}$.**
(PDF)

**S2 Appendix. Effect of finite sampling.**
(PDF)

**S3 Appendix. Relationship between the data means and correlations and the inferred couplings.**
(PDF)

## Author Contributions

**Conceptualization:** Valdemar Kargård Olsen, Yasser Roudi.

**Formal analysis:** Valdemar Kargård Olsen, Yasser Roudi.

**Software:** Valdemar Kargård Olsen.

**Validation:** Valdemar Kargård Olsen, Yasser Roudi.

**Visualization:** Valdemar Kargård Olsen.

**Writing – original draft:** Valdemar Kargård Olsen, Jonathan R. Whitlock, Yasser Roudi.

**Writing – review & editing:** Valdemar Kargård Olsen, Jonathan R. Whitlock, Yasser Roudi.

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
