## [Decision Letter · Decision Letter 0]

14 Dec 2023

Dear Dr. Roudi,

Thank you very much for submitting your manuscript "The quality and complexity of pairwise maximum entropy models for large cortical populations" for consideration at PLOS Computational Biology.

As with all papers reviewed by the journal, your manuscript was reviewed by members of the editorial board and by several independent reviewers. In light of the reviews (below this email), we would like to invite the resubmission of a significantly-revised version that takes into account the reviewers' comments.

We cannot make any decision about publication until we have seen the revised manuscript and your response to the reviewers' comments. Your revised manuscript is also likely to be sent to reviewers for further evaluation.

Sincerely,

Jonathan David Touboul

Academic Editor

PLOS Computational Biology

Daniele Marinazzo

Section Editor

PLOS Computational Biology

Reviewer's Responses to Questions

**Comments to the Authors:**

Reviewer #1: Olsen et al fit a large number of Ising models to Neuropixel recordings from multiple cortical areas of feely moving rats from a public dataset. They analyze how the quality of the model degrades with different features of the data such as population size, time binning or firing rates. The paper seems technically sound, and rigorous in varying different aspects of the analysis pipeline, which empirically expand theoretical results from Ref.10. It also introduces a couple of interesting statistical elements, in particular a new Ising partition function estimator and some lessons on the complex relationship between different empirical metrics of model quality used in practice. Experimentally we are not learning anything fundamentally new: it is a much more precise characterization of the long known observation that a simple Ising model only accounts for data well in small populations. Still, the manuscript does the grunt work of checking how this conclusion could be affected by various data and analysis considerations.

Detailed notes:

- I find dubious the scientific relevance of fitting a single Ising model to data that combines several computationally distinct sensory/behavioral conditions from multiple brain areas, but that concern is somewhat outside the scope of this paper. Still, I wonder if lumping together multiple conditions in this way also affects analysis in ways that DO matter for the goal at hand: naively one would think that combining subsets of data with different statistics (due to different computational regimes) results in mixture distributions with more complex associated pairwise correlation structure. Please discuss.

- The explanation of the core metrics needs substantial improvement. In particular I get the moment matching argument for the entropy difference definition of the KL, but it is sloppily done; similarly, the form of the factorized model is not terribly intuitive or properly explained. I imagine that a reader not already quite familiar with MaxEnt models would have a hard time following that. Either provide references together with each equation to make clear where it’s coming from, or (ideally and) explain it in an accessible way in the results.

-Similarly, while I appreciate the symmetry of having all quality measures ranging between 0 and 1 with zero ‘as good as factorized model’ and one ‘as good as actual data’, this involves additional nonstandard steps for some of the metrics and those need a little justification: why sqrt(l2 distance ratio) specifically? Please comment.

-What does “larger variability in firing rates” means precisely when discussing increases in G for the N<20 analysis? 1) the means of different neurons being more different, I.e. population heterogeneity, or 2) more variance in individual neural responses (which scales with the mean for both the factorized Bernoulli model or a more realistic Poisson firing )?

-Does the structure of the larger neural populations sub-factorize by area? If looking at the estimated Js in a mixes population including subsets of neurons from different brain regions then is that structure largely block diagonal, with weak across area interactions and stronger within area couplings?

-since the partition function estimator is critical for a good part of the results, its construction needs to be explained more clearly. I also don’t understand the logic of not describing its properties in the paper itself in a self-contained way.

- Critically, the results text is missing a clear explanation of how the entropy of the real data is being estimated for N>20; also how does one practically go from an estimate of Z to that for S_pair?

- Overall, across all analyses the critical variable affecting model quality is the product N\\bar\\nu\\delta t. The fundamental message of all analyses is to argue for a decrease in model performance as a function of this quantity. Given this agenda, one would need a little more initial motivation to justify why is this the critical quantity in the original theory and why should one expect it to be meaningful outside the perturbation regime of the original theory.

-I personally don’t find the SK subsection of the Results useful or experimentally relevant, despite the precedent of similar analyses in the retina. I also fundamentally question the interpretation of such metrics given that the data distribution was obtained by marginalizing across experimental conditions.

Minor:

- Section … is mentioned in the 2nd paragraph go Methods/Dataset , probably a legacy of a previous numbered section version of the text

-When introducing the constraints on the first and second moments (eq 1a,b) they are referred to as mean and correlations which is strictly speaking a little misleading.

-Title in Fig 2C is wrong, should be Different delta t

-Caption fig 3: nu bar typo

-Also not sure what the color is supposed to mean in 2G, and both panels in figure 3. All need meaningful color legends

DATA and CODE sharing: the data is public but the code for the results it is not, at least as far as i can see.

Reviewer #2: The manuscript describes quantification of maximum entropy models to fit statistics of recorded networks in the brain of a foraging rat. The recorded cells were distributed in several brain regions including visual, auditory, motor and somatosensory cortices. The authors perform analysis of the success of the PME models to predict several other statistics of networks’ dynamics. Overall, the analysis looks good and well performed. The authors also analyze predictions to larger populations and claim that the PME failed to predict the activity structure of these networks, a well know fact.

Reviewer #3: The review is uploaded as an attachment

**Have the authors made all data and (if applicable) computational code underlying the findings in their manuscript fully available?**

Reviewer #1: **No: **Data ok, missing github of code or similar.

Reviewer #2: Yes

Reviewer #3: Yes

PLOS authors have the option to publish the peer review history of their article (what does this mean?). If published, this will include your full peer review and any attached files.

Reviewer #1: No

Reviewer #2: No

Reviewer #3: **Yes: **Michele Castellana
---

## [Decision Letter · Decision Letter 1]

10 Apr 2024

Dear Dr. Roudi,

We are pleased to inform you that your manuscript 'The quality and complexity of pairwise maximum entropy models for large cortical populations' has been provisionally accepted for publication in PLOS Computational Biology.

Best regards,

Jonathan David Touboul

Academic Editor

PLOS Computational Biology

Daniele Marinazzo

Section Editor

PLOS Computational Biology

Reviewer's Responses to Questions

**Comments to the Authors:**

Reviewer #1: Thank you for addressing my concerns.

About point 14: the emphasis of that comment was on the fact that the second moment is related to 'correlations' but not mathematically the same thing.

take a random variable x,y and E denoting expectations under p(x), then

mean = E[x]

second moment = E[x^2]

variance = E[x^2] - E[x]^2

covariance = E[xy]-E[x]E[y]

correlation = (E[xy]-E[x]E[y]) / sqrt(Var(x)Var(y))

the constraint parametrized by J concerns the second moment, not correlations as such (although they are related and the literature is often sloppy and confounding the two)

**Have the authors made all data and (if applicable) computational code underlying the findings in their manuscript fully available?**

Reviewer #1: Yes

PLOS authors have the option to publish the peer review history of their article (what does this mean?). If published, this will include your full peer review and any attached files.

Reviewer #1: No

---

## [Editor Report · Acceptance letter]

23 Apr 2024

PCOMPBIOL-D-23-01436R1 

The quality and complexity of pairwise maximum entropy models for large cortical populations

Dear Dr Roudi,

I am pleased to inform you that your manuscript has been formally accepted for publication in PLOS Computational Biology. Your manuscript is now with our production department and you will be notified of the publication date in due course.

With kind regards,

Zsofia Freund
